# Tailoring interventions to suit self-reported format preference does not decrease vaccine hesitancy

Karl O. Mäki[1]*, Linda C. Karlsson[2], Johanna K. Kaakinen[1,3], Philipp Schmid[4], Stephan Lewandowsky[5,6], Jan Antfolk[7], Anna Soveri[2]

1 Department of Psychology and Speech-Language Pathology, University of Turku, Turku, Finland, 2 Department of Clinical Medicine, University of Turku, Turku, Finland, 3 INVEST Research Flagship, University of Turku, Turku, Finland, 4 Media and Communication Science, University of Erfurt, Erfurt, Germany, 5 School of Psychological Science, University of Bristol, Bristol, United Kingdom, 6 School of Psychological Science, University of Western Australia, Perth, Australia, 7 Department of Psychology, Åbo Akademi University, Turku, Finland

* otto.maki@utu.fi

**Data Availability Statement:** All data files are available from the OSF database (URL: osf.io/ubs26).

## Abstract

Individually tailored vaccine hesitancy interventions are considered auspicious for decreasing vaccine hesitancy. In two studies, we measured self-reported format preference for statistical vs. anecdotal information in vaccine hesitant individuals, and experimentally manipulated the format in which COVID-19 and influenza vaccine hesitancy interventions were presented (statistical vs. anecdotal). Regardless of whether people received interventions that were in line with their format preference, the interventions did not influence their vaccine attitudes or vaccination intentions. Instead, a stronger preference for anecdotal information was associated with perceiving the material in both the statistical and the anecdotal interventions as more frustrating, less relevant, and less helpful. However, even if the participants reacted negatively to both intervention formats, the reactions to the statistical interventions were consistently less negative. These results suggest that tailoring COVID-19 and influenza vaccine hesitancy interventions to suit people's format preference, might not be a viable tool for decreasing vaccine hesitancy. The results further imply that using statistics-only interventions with people who hold anti-vaccination attitudes may be a less risky choice than using only anecdotal testimonies.

## Introduction

Vaccine hesitancy is often used as an umbrella term for all types of attitudes and behaviors that question or go against vaccines [1]. It is a complex phenomenon that varies across different contexts and vaccines [2]. Efforts to decrease vaccine hesitancy often involve presenting the risks and benefits of vaccines in a statistical format [3–5]. In anti-vaccination messaging, on the other hand, emotional anecdotal testimonies of alleged vaccine adverse events are commonly used instead of, or in addition to, statistical information [6–10]. It is well known that

**Funding:** This project has received funding from the European Union's Horizon 2020 research and innovation program under grant agreement No 964728 (JITSUVAX). KOM was funded by the Faculty of Social Sciences at the University of Turku (www.utu.fi/en/university/faculty-of-social-sciences). JKK and KOM received funding from the Strategic Research Council's LITERACY program (Academy of Finland grant number: 335233). AS was funded by the Academy of Finland (grant number: 316004; www.aka.fi/en/). SL also acknowledges funding from the Humboldt Foundation Germany through a research award. The funders had no role in study design, data collection and analysis, decision to publish, or preparation of the manuscript.

**Competing interests:** The authors have declared that no competing interests exist.

anecdotal testimonies can affect people's health decisions and override statistical information in the decision-making process, although statistical information is more generalizable/informative when evaluating the risks and benefits of vaccines [11–15]. For example, in an experiment by Betsch et al. [11], participants were first presented with statistical information on the base rate of vaccine adverse events of a fictitious vaccine. After that, they were given ten personal stories from an online forum. The number of posts that included reports of vaccine adverse events varied between individuals. The results showed that the more narratives about vaccine adverse events the participants read, the lower their intention was to get vaccinated. Whereas statistical information is theorized to be persuasive because people perceive it as credible, verifiable, and generalizable [16], the persuasiveness of narratives has been attributed to their ability to reduce psychological reactance and counterarguing and to enable engagement in a storyline and identification with characters [17], while simultaneously being more easily processed than other types of formats [18]. The persuasive nature of narratives has led researchers to explore whether narratives can be utilized in interventions that promote vaccination. A recent systematic review [19] focusing on interventions and strategies aimed at decreasing parental vaccine hesitancy towards childhood vaccines in the U.S. identified storytelling, for example in the form of anecdotes, as an effective tool in decreasing vaccine hesitancy. The results of another recent systematic review [20] that investigated the effects of narrative interventions addressing vaccine hesitancy towards various kinds of vaccines, suggested that in most cases, anecdotal messages are slightly more effective than control messages in decreasing vaccine hesitancy, but the exact size of this effect remained inconclusive due to wide confidence intervals. When compared to educational and statistical interventions, the results were mixed, with some studies indicating that anecdotal interventions were more effective than statistical interventions, and others showing the opposite.

To the best of our knowledge, only one study has explored the effectiveness of narratives in reducing COVID-19 vaccine hesitancy [21]. The results from this between-subjects study comparing post-intervention responses in unvaccinated individuals showed that non-narrative texts (retrieved from the Centers for Disease Control and Prevention agency's website) were more effective than narratives and control texts in increasing confidence in COVID-19 vaccines. When it comes to vaccination intention, non-narrative texts, and self-persuasion narratives (i.e., narratives about an individual who has changed their mind about vaccination) were equally effective, and superior to control texts and narratives without self-persuasion. Taken together, although systematic reviews provide some, albeit weak, support for a benefit of using anecdotal material in vaccine hesitancy interventions, there is still considerable heterogeneity between individual studies.

One way to better understand the sometimes-ambiguous findings regarding the benefits of anecdotes in interventions, is to explore individual differences that may affect the intervention outcome. Previous studies suggest that health interventions [22], and vaccine hesitancy interventions [23] are more effective when they are tailored to fit the needs of the recipient. Designing tailored interventions is believed to increase recipients' attention to the message and by extension to make the message easier to remember. Tailored interventions are also expected to more often be perceived as relevant and to elicit more thorough elaboration of tailored messages compared to non-tailored ones [24]. Considering these findings, that 1) the research on the effectiveness of statistical and anecdotal vaccine hesitancy interventions is mixed, and 2) that tailored interventions tend to be superior to one-size-fits-all approaches, we explored whether the efficacy of anecdotal interventions is dependent on the degree to which people prefer to base their health decisions on other people's experiences (anecdotes) over statistics, or in other words, whether it would be beneficial to tailor vaccine hesitancy interventions according to the recipient's format preference. In the present study, we created statistical and

anecdotal intervention materials to promote both COVID-19 and influenza vaccines in order to increase the generalizability of the results. In two randomized controlled experiments, we tested whether the efficacy of the interventions was associated with peoples' self-reported format preference. Only individuals who before the intervention reported some degree of hesitancy towards getting vaccinated were included in the study. To the best of our knowledge, this is the first study to address the issue of tailoring vaccine hesitancy interventions according to format preference. Despite the fact that this study was exploratory in nature, we hypothesized based on previous information format and message tailoring research that the interventions would be more efficient in reducing vaccine hesitancy when the intervention format was congruent with the individuals' format preference than when it was incongruent.

We furthermore investigated whether recipients' format preference was related to how the messages were received by the participants, that is, to what degree they experienced the information in the interventions to be relevant, helpful, and/or frustrating. Since previous research has found tailored health interventions to be perceived as more relevant than non-tailored one's [24,25], we also hypothesized that the tailored interventions would be perceived as more helpful than the non-tailored interventions. Moreover, while studies have shown that attempts to affect people's attitudes and choices might cause frustration [26]—or at worst even backfire and increase vaccine hesitancy in people who already hold very negative attitudes to vaccines [27,28]—we hypothesized that people would react more positively to interventions that are presented in the format (statistical vs. anecdotal) that they prefer [29]. These results would be important, as they too would provide evidence for the benefits of tailored interventions. In summary, our hypotheses were as follows:

1. Statistical and anecdotal COVID-19 and influenza vaccine hesitancy interventions increase participants' vaccination intentions and pro-vaccination attitudes as compared to the control group.

2. Format preference is associated with the efficacy of the anecdotal vaccine hesitancy interventions so that a stronger preference for anecdotes is related to a greater pre–post intervention increase in participants' vaccination intentions and pro-vaccination attitudes, whereas a stronger preference for statistics is associated with a greater pre–post intervention increase in participants' vaccination intentions and pro-vaccination attitudes in the statistical vaccine hesitancy interventions.

3. Intervention reception is related to the intervention format and participants' format preference so that format preference congruent interventions elicit less frustration and are perceived as more relevant and helpful than non-congruent interventions.

### COVID-19 and influenza vaccinations in the Finnish context

In Finland, COVID-19 vaccinations have been available and recommended to the general public since the beginning of 2021. At the time of writing, around 90% of the adult Finnish population has received two COVID-19 vaccine doses and approximately 60% has received a third dose [30]. The Finnish Institute for Health and Welfare (THL) recommends three COVID-19 vaccine doses for all adults (18+ years) and a fourth dose for people with severe immunodeficiencies [31,32]. Throughout the pandemic, COVID-19 vaccinations have been free of charge and voluntary for the general public. Proof of vaccination has, however, been a requirement for traveling and to some extent for visiting places such as bars and restaurants. COVID-19 vaccinations are currently (until the end of 2022) mandatory for healthcare professionals who work in close contact with patients belonging to risk populations [33].

Influenza vaccines are voluntary and typically not free of charge for the general population. However, the influenza vaccines are recommended to people who regularly come into close contact with risk groups and are free of charge for example for risk groups, children under the age of seven, and people living in institutional conditions. For healthcare professionals working with at-risk patients, the influenza vaccine has been mandatory since March 2017 [34]. Some employers also offer free influenza vaccinations to their staff. In Finland, the influenza vaccination coverage was around 40% in 2020–2021 [35].

## Method

In two randomized controlled experiments (one on COVID-19 and one on influenza) we explored whether peoples' self-reported format preference predicted the efficacy of statistical and anecdotal interventions. Both experiments were pre-registered at OSF (link: osf.io/jqspe).

### Ethics statement

Ethical permission for all studies was given by the Research Ethics Committee in Psychology and Logopedics of Åbo Akademi University. Before participating in the experiments, all participants were presented with information about the study and about the management of the data. Participants were then asked to indicate that they were at least 18 years old and that they had received enough information about the study to be able to give their informed consent to participate. All participants gave their written informed consent by ticking a box. Participants were also informed that participation was voluntary and that withdrawal from the study was possible at all times.

### Participants and procedure

In October 2021, an electronic survey was advertised on Facebook, Messenger, and Instagram to adults living in Finland. This was considered an efficient and cost-effective way to collect data from the Finnish population. In 2021, Finnish Facebook users were estimated to account for approximately 63.5% of the total population, with age groups between 18 and 65+ being well represented [36]. When entering the survey, the respondents were first asked to give their informed consent. After that, they filled out a short survey measuring trust in health authorities, general trust in vaccines, conspiracy mentality, and vaccination behavior. These variables will be analyzed and published as part of another study.

After this, the participants were randomly presented with either a question about how likely they were to take a seasonal COVID-19 vaccine, should that become relevant, or about how likely they were to take the next seasonal influenza vaccine. These questions were used to assess the eligibility of respondents. To decrease the risk of ceiling-effects and to avoid unnecessary interventions, we determined that an appropriate inclusion criterion was that participants would have to be less than 80% likely to get vaccinated. This decision was made prior to the data collection. Those who met the inclusion criteria proceeded to the intervention related to that particular vaccine. Those who stated that it was at least 80% likely that they would take the vaccine in question, were given the question regarding the other vaccine. We calculated the statistical power of our analyses using data simulation. We manipulated the number of observations to find a large enough sample size for which a multiple regression model would reliably detect significant interaction terms between format preference and intervention groups. We found that a sample size of 600 participants per experiment would yield a statistical power of 0.87 for weak associations (three-point change on a scale from 0–100) between format preference and the efficacy of statistical and anecdotal interventions (see the S1 File for an overview of the power simulation). Altogether 1942 people responded to one or both of the eligibility

questions. Of these, 843 respondents were excluded because they were at least 80% likely to take both vaccines. The final sample, thus, consisted of 1099 respondents. Of those, 559 took part in the COVID-19 experiment and 540 in the influenza experiment. For both sample demographics, see S1 Table in S2 File.

The participants in the COVID-19 experiment first reported their likelihood of getting a third COVID-19 vaccine and answered statements probing COVID-19 vaccine attitudes. They were then randomized into one of the three following groups: 1) statistical intervention, 2) anecdotal intervention, or 3) control. The participants in the statistical and anecdotal groups were presented with texts about COVID-19 vaccines given in either a statistical or anecdotal format depending on their intervention group. The control group received information on a topic unrelated to vaccines. After reading the material, the participants' level of frustration was measured. Next, they answered the same vaccination intention questions as before the intervention. They were also asked how the intervention material had affected their vaccination intentions and how relevant and helpful they found the intervention materials to be. Following this, they were administered the same vaccine-attitude statements as before the intervention. The format preference scale appeared randomly either directly before the intervention material or at the very end of the survey to counterbalance a possible order effect. The procedure was the same for the participants in the influenza experiment. For an overview of the procedure, see S1 Fig in S2 File.

## Materials

**Intervention materials.** In both experiments, and for all groups, the intervention materials consisted of three short texts. For the statistical and anecdotal groups, these texts were designed to give information about the threat of the disease and the safety and benefit of the vaccine, as well as about the importance of protecting others through vaccination. For those in the statistical group, the information was derived from THL's webpage and presented in a statistical format (e.g., the rate at which the coronavirus spreads). In the anecdotal group, the material was in an anecdotal format (e.g., a personal story about having received the vaccine). These anecdotes were carefully crafted out of real anecdotes people have posted online on sites such as www.voicesforvaccines.org. The control group received similar short texts that were unrelated to vaccines and consisted of made-up anecdotal experiences about a made-up musical institute. Fig 1 shows English translations of the COVID-19 intervention materials as an example. See S2–S6 Figs and S2–S6 Tables in S2 File for all intervention materials and English translations.

**Pre- and posttest measures.** Measures of vaccination intentions and vaccine attitudes were administered before and after the intervention to assess intervention efficacy. These measures were created for this study. The participants responded to all pre- and posttest measures on a slider scale where the numerical anchors (0–100) were not visible. *Vaccination intentions* were measured with two questions in the COVID-19 experiment ("How likely would you be to take a seasonal COVID-19 vaccine, should one become available?" and "How likely would you be to take a third COVID-19 vaccine, if it was recommended by the health authorities?") and one question in the influenza experiment ("How likely are you to take the next seasonal influenza vaccine?"). The vaccination intention questions were measured on a scale ranging from 0 (very unlikely) to 100 (very likely). In each experiment, five questions were used to measure *vaccine attitudes*: 1) perceived vaccination effort was measured with the question "How easy is it to get the [vaccine] in Finland?" (0 = not easy at all; 100 = very easy); 2) perceived disease threat was measured with the question "How big of a threat is [disease] to your health?" (0 = not threatening at all; 100 = very threatening); 3) perceived vaccine safety was measured

**Statistical intervention**

**Anecdotal intervention**

**Control material**

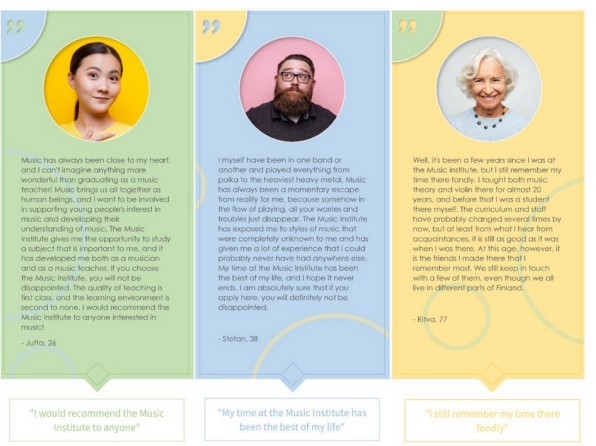

**Fig 1. COVID-19 intervention materials (English translations).**

with the question "How safe do you think that the [vaccine] is/are?" (0 = not safe at all; 100 = very safe); 4) perceived vaccine efficacy was measured with the question "How efficient do you think that the [vaccine] is/are?" (0 = not efficient at all; 100 = very efficient); and 5) vaccination altruism was measured with the statement "It is important to take the [vaccine] as it also protects others." (0 = strongly disagree; 100 = strongly agree).

**Format preference.** We conducted three studies with the aim to develop and validate a scale to measure an individual's self-reported preference for receiving information in either a statistical or an anecdotal format when making health-related decisions. These studies are reported in the S3 File. The three studies resulted in the 6-item scale presented in Table 1. The scale demonstrated good reliability and was empirically associated with theoretically related factors in the expected way, so that a stronger preference for anecdotal information was related to lower trust in vaccines, lower trust in health authorities, and stronger conspiracy mentality. The scale was also partly able to predict how susceptible the individual was to the influence of anecdotes. Furthermore, the results showed that negative vaccine anecdotes had a stronger and a more consistent influence on vaccine attitudes and vaccination intentions than positive vaccine anecdotes. As the scale indicated good reliability and validity, we used it to measure format preference in this study.

**Table 1. The format preference scale.**

| | Item |
|---|---|
| 1. | I mostly make decisions about my health based on the statistical information available.* |
| 2. | I think people's first-hand experiences tell me more about the safety of a medical procedure than statistical research results. |
| 3. | It is easier for me to make decisions about my health based on other people's experiences than on statistical information. |
| 4. | Individual people's experiences have a big influence on my health-related decisions. |
| 5. | When faced with statistical data that contradicts people's experiences, I prefer to trust people's reported experiences. |
| 6. | I find it easier to make health decisions based on statistical information than on other people's experiences.* |

Response scale: 1 = strongly disagree to 7 = strongly agree.

*Reversed item.

**Intervention reception.** Three Likert scale measures were created and used in this study ssssto assess the participants' reactions to the intervention materials: 1) *frustration* was measured with the question "Did you get irritated while reading the texts?" (1 = not at all to 7 = very much); 2) *message relevance* was measured with the question "How relevant were the contents of the texts for you?" (1 = completely irrelevant to 7 = very relevant); and 3) *message helpfulness* was measured with the question "How helpful were the contents of the texts for you?" (1 = completely useless to 7 = very helpful).

**Self-reported influence of intervention.** Self-reported influence of the interventions was measured with the question "How did the texts you read influence your intention to take the [vaccine]?" (1 = significantly decreased my intention to take the vaccine; 2 = somewhat decreased my intention to take the vaccine; 3 = did not affect my intention to take the vaccine; 4 = somewhat increased my intention to take the vaccine; 5 = significantly increased my intention to take the vaccine).

## Statistical analyses

The data from the two experiments were analyzed separately, but the same statistical procedure was followed in both cases. The analyses were conducted in R version 4.0.5 [37].

**Preliminary analyses.** Our aim was to use composite scores for the vaccine attitudes construct (five items) and the format preference scale (six items). We therefore first used confirmatory factor analysis (CFA) to investigate whether the items of each construct loaded on their respective factor. Moreover, we calculated Cronbach's alpha to evaluate the internal consistency of the format preference scale. We also ran separate ANOVA models with each pretest score as a dependent variable and with the intervention group as the independent variable to confirm that the pretest measure scores were balanced between the three intervention groups.

**Efficacy of interventions.** We conducted multiple regression analyses to investigate whether the interventions were more effective in decreasing vaccine hesitancy when the intervention format aligned with the format people reported that they preferred. Due to responses being strongly skewed and even slightly bimodal at times, we used change scores for the main analyses. Thus, the resulting outcome measures were the change in vaccination intentions and vaccine attitudes between pre- and posttest. The change scores were created by subtracting pretest scores from posttest scores. Positive change scores thus indicated higher vaccination intention and more positive vaccine attitudes after the intervention. A sum score was calculated for the format preference item, so that higher scores indicated a stronger preference for

anecdotal information. Separate regression analyses were conducted for each outcome. The predictors in all analyses were intervention group and format preference, including their interaction term. Treatment contrasts were applied, and the control group was set as the baseline to which the statistical and the anecdotal groups were compared. The format preference variable was centered and scaled around the mean. These regression analyses gave us two types of information. First, the main effect of intervention (statistical vs. control and anecdotal vs. control) can be interpreted as the effectiveness of that intervention if it were administered in a non-tailored manner in a population where format preference is evenly distributed. Second, the interaction effects (intervention by format preference) can be interpreted as the effect if interventions were tailored to fit the preference of the respondent. For example, the interaction results would support the hypothesis of the benefit of tailored interventions if individuals with *higher* scores on the format preference scale (i.e., preference for anecdotes) have a greater decrease in vaccine hesitancy in the *anecdotal* intervention group than in the statistical intervention. Conversely, in the *statistical* intervention group, the hypothesis of the benefit of tailored interventions would be supported if individuals with *lower* scores on the format preference scale (i.e., preference for statistics) have a greater decrease in vaccine hesitancy. Format preference should be unrelated to the change in vaccine hesitancy in the control group. We investigated significant interactions by setting the intervention group of interest as the reference and repeating the analysis. This way we were able to get the simple effects of format preference on the outcome variables for specific intervention groups.

**Intervention reception.** To explore how format preference and intervention format related to how the respondents experienced the intervention formats, we tested three multiple regression models with frustration, message relevance, and message helpfulness respectively as the outcomes. The predictors were again intervention group (statistical vs. control and anecdotal vs. control) and format preference, including their interaction term. Because we were also interested in the difference between the statistical group and the anecdotal group in intervention reception, we reran these regression models with the statistical group as the reference group.

## Results

### The COVID-19 experiment

**Preliminary analyses.** The fit indices from the CFA for the COVID-19 vaccine attitude factor suggested good fit (Table 2). However, the factor loading of the vaccination-effort item on the COVID-19 vaccine attitudes factor was negligible (S7 Table in S2 File). A follow-up CFA without the vaccination-effort item resulted in a mixed fit. Hence, we did not create a composite of the five vaccine attitude items but ran the multiple regression analyses with the vaccination effort, disease threat, and vaccination altruism items separately, and the vaccine safety and vaccine efficacy items as a combined variable (hereafter, *vaccine confidence*). We based this division on the 5C model [38]. The resulting variables matched four of the 5C

**Table 2. Goodness-of-fit-indicators (pretest measures): The COVID-19 experiment.**

| Model | $\chi^2$ | df | CFI | TLI | SRMR | RMSEA |
|---|---|---|---|---|---|---|
| Vaccine attitudes (all items) | 13.94 | 5 | .99 | .98 | .02 | .06 |
| Vaccine attitudes (4 items) | 31.14 | 2 | .97 | .90 | .03 | .16 |
| Format preference | 96.63 | 8 | .99 | .97 | .04 | .14 |

CFI > .90 = good fit; TLI > .90 = good fit; SRMR < .08 = good fit; RMSEA < .08 = good fit.

**Table 3. Vaccination intention and attitude mean scores by intervention group: The COVID-19 experiment.**

| Variable | Statistical | | Anecdotal | | Control | |
|---|---|---|---|---|---|---|
| | *Mean* | *SD* | *Mean* | *SD* | *Mean* | *SD* |
| **Pretest** | | | | | | |
| Seasonal vaccine | 22.89 | 28.01 | 19.73 | 28.20 | 18.17 | 27.61 |
| Third dose | 21.95 | 32.05 | 20.26 | 32.41 | 18.49 | 31.31 |
| Vaccination effort | 85.20 | 20.50 | 88.46 | 17.92 | 86.23 | 20.02 |
| Vaccine confidence | 33.35 | 29.70 | 27.56 | 30.47 | 26.31 | 30.69 |
| Disease threat | 30.34 | 28.07 | 29.13 | 30.06 | 28.61 | 29.50 |
| Vaccination altruism | 30.49 | 35.28 | 28.73 | 37.58 | 27.04 | 35.50 |
| **Posttest** | | | | | | |
| Seasonal vaccine | 23.15 | 31.40 | 21.09 | 31.34 | 19.38 | 30.79 |
| Third dose | 23.66 | 33.73 | 21.36 | 33.56 | 19.81 | 32.29 |
| Vaccination effort | 85.06 | 20.63 | 88.28 | 17.85 | 86.76 | 20.08 |
| Vaccine confidence | 33.04 | 31.29 | 27.11 | 31.52 | 25.76 | 31.30 |
| Disease threat | 30.54 | 28.98 | 29.21 | 30.73 | 26.01 | 28.83 |
| Vaccination altruism | 30.70 | 35.47 | 28.68 | 37.12 | 25.90 | 35.31 |
| **Change score** | | | | | | |
| Seasonal vaccine | 0.25 | 16.70 | 1.36 | 13.11 | 1.21 | 10.10 |
| Third dose | 1.71 | 11.43 | 1.10 | 7.57 | 1.32 | 5.43 |
| Vaccination effort | -0.14 | 6.20 | -0.18 | 6.75 | 0.52 | 7.15 |
| Vaccine confidence | -0.31 | 9.42 | -0.45 | 7.87 | -0.55 | 6.71 |
| Disease threat | 0.20 | 7.89 | 0.08 | 14.30 | -2.60 | 14.40 |
| Vaccination altruism | 0.21 | 6.52 | -0.05 | 7.21 | -1.14 | 10.42 |

Statistical = statistical group; Anecdotal = anecdotal group; Control = control group; Seasonal vaccine = intention to take a seasonal COVID-19 vaccine (change score); Third dose = intention to take a third COVID-19 vaccine dose (change score); Vaccination effort = perceived effort of getting a COVID-19 vaccine (change score); Vaccine confidence = confidence in the COVID-19 vaccine (change score); Disease threat = perceived threat of COVID-19 (change score); Vaccination altruism = perceived importance of getting a COVID-19 vaccine to protect others (change score).

model's psychological antecedents of vaccine hesitancy; 1) constraints, 2) complacency, 3) collective responsibility, and 4) confidence. Vaccine confidence was calculated as the mean of the vaccine safety and efficacy items. Response distributions for the vaccine attitudes items can be found in S7 Fig in S2 File. For the format preference scale, all fit indices, except the RMSEA, suggested excellent fit and Cronbach's alpha was good (Cronbach's α = .87). A composite score of the format preference scale was thus used in subsequent analyses. Response distributions for the format preference items are presented in S8 Fig in S2 File. The ANOVA models showed that the pretest scores did not statistically significantly differ between the groups, indicating that the groups were balanced before the interventions (see S8 Table in S2 File).

**Efficacy of non-tailored interventions.** For all pre- and posttest scores, as well as change scores, see Table 3. For the full multiple regression models, see Table 4. From the results of the multiple regression analyses, we first report the main effects of the interventions, which gives information on the efficacy of the statistical and anecdotal interventions if they were administered in a non-tailored way. The results showed that there were no statistically significant differences between the control group and the statistical group, or the control group and the anecdotal group, when it comes to the change in COVID-19 vaccination intentions, perceived effort to get a vaccine, vaccine confidence, or vaccination altruism. There was, however, a statistically significant difference in perceived COVID-19 disease threat between the control

**Table 4. Results from the multiple regression analyses on the effects of intervention and format preference: The COVID-19 experiment.**

| Outcome | Predictor | b | 95% CI | p |
|---|---|---|---|---|
| Seasonal vaccine | Intercept (Control) | 1.13 | -0.73, 2.99 | .233 |
| | Statistical | -0.90 | -3.88, 2.07 | .550 |
| | Anecdotal | 0.27 | -2.21, 2.76 | .828 |
| | FP | -1.60 | -3.50, 0.30 | .098 |
| | FP*Statistical | -0.57 | -3.59, 2.45 | .711 |
| | FP*Anecdotal | -0.19 | -2.68, 2.31 | .883 |
| Third dose | **Intercept (Control)** | **1.31** | **0.17, 2.45** | **.025** |
| | Statistical | 0.31 | -1.51, 2.13 | .738 |
| | Anecdotal | -0.19 | -1.72, 1.33 | .802 |
| | FP | -0.41 | -1.57, 0.76 | .492 |
| | **FP*Statistical** | **1.97** | **0.12, 3.82** | **.037** |
| | FP*Anecdotal | -0.19 | -2.68, 2.31 | .883 |
| Vaccination effort | Intercept (Control) | 0.57 | -0.40, 1.54 | .247 |
| | Statistical | -0.68 | -2.23, 0.87 | .390 |
| | Anecdotal | -0.80 | -2.09, 0.50 | .226 |
| | FP | 0.81 | -0.18, 1.80 | .107 |
| | FP*Statistical | -1.14 | -2.71, 0.43 | .155 |
| | FP*Anecdotal | -0.32 | -1.62, 0.98 | .633 |
| Vaccine confidence | Intercept (Control) | -0.57 | -1.69, 0.55 | .317 |
| | Statistical | 0.16 | -1.64, 1.95 | .864 |
| | Anecdotal | 0.19 | -1.31, 1.69 | .807 |
| | FP | -0.24 | -1.39, 0.90 | .676 |
| | FP*Statistical | 0.18 | -1.64, 2.00 | .844 |
| | FP*Anecdotal | -0.67 | -2.18, 0.84 | .382 |
| Disease threat | **Intercept (Control)** | **-2.72** | **-4.61, -0.83** | **.005** |
| | Statistical | 2.93 | -0.09, 5.96 | .057 |
| | Anecdotal | 2.74 | 0.21, 5.26 | .034 |
| | FP | -1.06 | -2.99, 0.87 | .282 |
| | FP*Statistical | 0.93 | -2.14, 4.00 | .551 |
| | FP*Anecdotal | 0.81 | -1.73, 3.35 | .531 |
| Vaccination altruism | Intercept (Control) | -1.11 | -2.30, 0.09 | .069 |
| | Statistical | 1.29 | -0.62, 3.19 | .186 |
| | Anecdotal | 1.04 | -0.56, 2.63 | .201 |
| | FP | 0.50 | -0.72, 1.72 | .421 |
| | FP*Statistical | 0.24 | -1.70, 2.18 | .808 |
| | FP*Anecdotal | -0.36 | -1.96, 1.24 | .657 |
| Frustration | **Intercept (Control)** | **2.12** | **1.86, 2.39** | **< .001** |
| | **Statistical** | **1.21** | **0.78, 1.63** | **< .001** |
| | **Anecdotal** | **1.12** | **0.76, 1.47** | **< .001** |
| | FP | -0.09 | -0.36, 0.18 | .532 |
| | FP*Statistical | 0.37 | -0.06, 0.80 | .088 |
| | **FP*Anecdotal** | **0.81** | **0.45, 1.16** | **< .001** |
| Message relevance | **Intercept (Control)** | **2.23** | **1.99, 2.46** | **< .001** |
| | **Statistical** | **1.30** | **0.92, 1.67** | **< .001** |
| | **Anecdotal** | **0.60** | **0.28, 0.91** | **< .001** |
| | FP | 0.11 | -0.13, 0.35 | .387 |
| | FP*Statistical | -0.41 | -0.79, -0.02 | .038 |

(*Continued*)

**Table 4.** (Continued)

| Outcome | Predictor | b | 95% CI | p |
|---|---|---|---|---|
| | **FP*Anecdotal** | **-0.47** | **-0.78, -0.15** | **.004** |
| Message helpfulness | **Intercept (Control)** | **1.96** | **1.74, 2.17** | **< .001** |
| | **Statistical** | **1.21** | **0.87, 1.56** | **< .001** |
| | **Anecdotal** | **0.41** | **0.13, 0.70** | **.005** |
| | FP | 0.05 | -0.17, 0.27 | .640 |
| | **FP*Statistical** | **-0.37** | **-0.72, -0.02** | **.036** |
| | **FP*Anecdotal** | **-0.45** | **-0.74, -0.17** | **.002** |

Statistical = statistical group; Anecdotal = anecdotal group; Control = control group; FP = format preference; Seasonal vaccine = intention to take a seasonal COVID-19 vaccine (change score); Third dose = intention to take a third COVID-19 vaccine dose (change score); Vaccination effort = perceived effort of getting a COVID-19 vaccine (change score); Vaccine confidence = confidence in the COVID-19 vaccine (change score); Disease threat = perceived threat of COVID-19 (change score); Vaccination altruism = perceived importance of getting a COVID-19 vaccine to protect others (change score); Frustration = frustration caused by the intervention; Message relevance = perceived relevance of the intervention material; Message helpfulness = perceived helpfulness of the intervention material;

* = interaction between predictors;

bold = statistically significant effect.

group and the anecdotal group, $b = 2.74$, 95% CI[0.21, 5.26], $p = .034$, stemming from the fact that the perceived threat of COVID-19 decreased in the control group, but not in the anecdotal group. The difference between the control group and the statistical group on this outcome was not statistically significant. Taken together, these results indicate that statistical and anecdotal interventions would not decrease vaccine hesitancy if the interventions were administered in a non-tailored manner.

**Efficacy of tailored interventions.** Because treatment contrasts were applied, the main effects of format preference represent the associations between format preference and the outcomes in the control group. None of these associations were statistically significant. The results concerning the interaction effects—which provide information on the efficacy of tailoring interventions according to the recipients' format preference—showed that there were no statistically significant interactions between format preference and intervention group when it comes to pre-post changes in intention to take a seasonal COVID-19 vaccine, vaccine confidence, perceived disease threat, or vaccination altruism. There was, however, a significant interaction indicating that the relationship between format preference and the change in the intention to take a third COVID-19 vaccine in the statistical group differed from that in the control group, $b = 1.97$, CI[0.12, 3.82], $p = .037$. More specifically, the simple effect of format preference on the intention to take a third COVID-19 vaccine was positive and significant in the statistical group, $b = 1.56$, CI[0.12, 3.00], $p = .034$. This result was in the unexpected direction, as it means that the more participants preferred anecdotes, the more their intentions to take a third COVID-19 vaccine increased by the statistical intervention. The corresponding interaction for the comparison between the anecdotal and the control group was not significant. Taken together, these results suggested that the statistical and anecdotal interventions would not be effective even if the recipients had been given the intervention in the format they preferred.

**Intervention reception.** Next, we explored how the participants experienced the interventions. The multiple regression analyses revealed that the participants became significantly more frustrated from reading the statistical and anecdotal intervention materials, compared to the control material, $b = 1.21$, CI[0.78, 1.63], $p < .001$, and, $b = 1.12$, CI[0.76, 1.47], $p < .001$, respectively. There was no significant difference in frustration between the statistical and the anecdotal interventions.

**Table 5. Self-reported influence of the interventions: The COVID-19 experiment.**

| Response | Statistical | Anecdotal | Control |
|---|---|---|---|
| Somewhat or significantly increased intentions | 3.3% | 2.9% | 1.6% |
| Somewhat or significantly decreased intentions | 27.6% | 23.0% | 8.3% |
| Had no effect on intentions | 69.1% | 74.1% | 90.1% |

Statistical = statistical group; Anecdotal = anecdotal group; Control = control group.

The statistical and anecdotal intervention materials were, however, perceived as more relevant compared to the control group, $b$ = 1.30, CI[0.92, 1.67], $p < .001$, and, $b$ = 0.60, CI[0.28, 0.91], $p < .001$, respectively. The statistical intervention was furthermore perceived as more relevant than the anecdotal intervention, $b$ = -0.70, CI[-1.06, -0.34], $p < .001$. The intervention material was also perceived as more helpful in the statistical and anecdotal groups, than in the control group, $b$ = 1.21, CI[0.87, 1.56], $p < .001$, and, $b$ = 0.41, CI[0.13, 0.70], $p = .005$, respectively. The statistical intervention was considered more helpful than the anecdotal intervention, $b$ = -0.80, CI[-1.13, -0.47], $p < .001$.

There were no statistically significant associations between format preference and frustration, message relevance or message helpfulness in the control group. The association between format preference and frustration differed between the anecdotal group and the control group, as indicated by the statistically significant interaction, $b$ = 0.81, CI[0.45, 1.16], $p < .001$. The corresponding interaction in the statistical group was not statistically significant. The simple effect of format preference on frustration in the anecdotal group was positive and statistically significant, $b$ = 0.72, CI[0.49, 0.95], $p < .001$, suggesting that the more an individual preferred anecdotes, the more frustrated they became from the anecdotal intervention. The interaction effects further indicated that the relationship between format preference and the perceived relevance, as well as the perceived helpfulness of the interventions, differed between the control group and the statistical group ($b$ = -0.41, CI[-0.79, -0.02], $p = .038$, and, $b$ = -0.37, CI[-0.72, -0.02], $p = .036$, respectively) and the control group and the anecdotal group ($b$ = -0.47, CI[-0.78, -0.15], $p = .004$, and, $b$ = -0.45, CI[-0.74, -0.17], $p = .002$, respectively). The simple effects of format preference on perceived relevance and perceived helpfulness of the interventions were negative and statistically significant in both the statistical ($b$ = -0.30, CI[-0.60, -0.00], $p = .048$, and, $b$ = -0.32, CI[-0.59, -0.05], $p = .020$) and the anecdotal group ($b$ = -0.36, CI[-0.56, -0.16], $p = .001$, and, $b$ = -0.40, CI[-0.59, -0.21], $p < .001$), indicating that the more a person preferred anecdotes the less relevant and helpful they found the interventions to be. These results suggest that individuals who had a stronger preference for anecdotes over statistics tended to consider the interventions as less relevant and less helpful regardless of which format the intervention information was presented in.

**Self-reported influence of the interventions.** Finally, although the majority of the participants in the statistical and anecdotal groups stated that the interventions did not change their intentions to get vaccinated against COVID-19, around 25% of participants in both groups reported that the interventions *decreased* their intentions to get vaccinated against COVID-19, whereas only 3% stated that the intervention had increased their intention. See Table 5 for the self-reported influence of the interventions.

## The influenza experiment

**Preliminary analyses.** All fit indices, except the RMSEA, suggested good fit for the influenza vaccine attitude items (Table 6). However, the vaccination effort item did not load on the

**Table 6. Goodness-of-fit indicators (pretest measures): The influenza experiment.**

| Model | $\chi^2$ | df | CFI | TLI | SRMR | RMSEA |
|---|---|---|---|---|---|---|
| Vaccine attitudes (all items) | 0.00 | 5 | .97 | .94 | .04 | .10 |
| Vaccine attitudes (4 items) | 0.00 | 2 | .94 | .83 | .05 | .22 |
| Format preference | 0.00 | 8 | .99 | .98 | .03 | .15 |

CFI > .90 = good fit; TLI > .90 = good fit; SRMR < .08 = good fit; RMSEA < .08 = good fit.

influenza vaccine attitudes factor (S9 Table in S2 File). A follow-up CFA without the vaccination-effort item resulted in a mixed fit. Thus, as in the COVID-19 experiment, we conducted the multiple regression analyses with the vaccination effort, disease threat, and vaccination altruism items separately and combined the vaccine safety and vaccine efficacy items to the aggregate item *vaccine confidence*. The vaccine confidence item was again calculated as the mean of the two items. Response distributions for the vaccine attitudes items can be found in S9 Fig in S2 File. Again, all fit indices, except the RMSEA, suggested excellent fit for the format preference items, and the internal consistency of the scale was good (Cronbach's α = .89). Response distributions for the format preference items are presented in S10 Fig in S2 File. The pretest scores did not statistically significantly differ between the groups according to the ANOVA models, indicating that the groups were balanced before the interventions (see S10 Table in S2 File).

**Efficacy of non-tailored interventions.** For the pre- and posttest scores, and the change scores, see Table 7. See Table 8 for the full multiple regression models. The multiple regression

**Table 7. Vaccination intention and attitude mean scores by intervention group: The influenza experiment.**

| Variable | Statistical | | Anecdotal | | Control | |
|---|---|---|---|---|---|---|
| | *Mean* | *SD* | *Mean* | *SD* | *Mean* | *SD* |
| **Pretest** | | | | | | |
| Vaccination intention | 24.12 | 27.57 | 20.74 | 27.49 | 19.34 | 26.59 |
| Vaccination effort | 80.68 | 21.72 | 82.64 | 19.16 | 82.13 | 20.87 |
| Vaccine confidence | 58.36 | 30.39 | 57.70 | 30.84 | 54.83 | 30.72 |
| Disease threat | 23.59 | 20.63 | 23.53 | 23.31 | 23.69 | 23.44 |
| Vaccination altruism | 38.69 | 32.51 | 38.79 | 32.13 | 36.76 | 33.32 |
| **Posttest** | | | | | | |
| Vaccination intention | 30.12 | 30.80 | 26.32 | 31.02 | 22.85 | 27.89 |
| Vaccination effort | 80.30 | 20.73 | 81.40 | 20.36 | 81.58 | 20.24 |
| Vaccine confidence | 58.06 | 31.82 | 56.42 | 32.09 | 52.61 | 32.10 |
| Disease threat | 23.19 | 22.37 | 23.62 | 24.65 | 23.87 | 24.70 |
| Vaccination altruism | 43.69 | 35.25 | 41.01 | 33.70 | 37.44 | 34.04 |
| **Change score** | | | | | | |
| Vaccination intention | 6.00 | 16.04 | 5.58 | 15.69 | 3.51 | 13.51 |
| Vaccination effort | -0.38 | 10.84 | -1.24 | 11.64 | -0.55 | 8.29 |
| Vaccine confidence | -0.30 | 8.99 | -1.28 | 8.39 | -2.22 | 10.83 |
| Disease threat | -0.40 | 10.20 | 0.10 | 11.80 | 0.18 | 11.99 |
| Vaccination altruism | 5.00 | 15.55 | 2.22 | 11.54 | 0.68 | 6.67 |

Statistical = statistical group; Anecdotes = anecdotal group; Control = control group; Next vaccine = intention to take the next seasonal influenza vaccine; Vaccination effort = perceived effort of getting the influenza vaccine; Vaccine safety = perceived safety of the influenza vaccine; Vaccine efficacy = perceived efficacy of the influenza vaccine; Disease threat = perceived threat of influenza; Vaccination altruism = perceived importance of getting the influenza vaccine to protect others.

**Table 8. Results from the multiple regression analyses on the effects of intervention and format preference: The influenza experiment.**

| Outcome | Predictor | *b* | 95% CI | *p* |
|---|---|---|---|---|
| Next vaccine | **Intercept (Control)** | **3.55** | **1.42, 5.67** | **.001** |
| | Statistical | 2.42 | -0.98, 5.82 | .163 |
| | Anecdotal | 2.12 | -0.77, 5.00 | .150 |
| | FP | 0.10 | -2.17, 2.38 | .928 |
| | FP*Statistical | -3.12 | -6.49, 0.25 | .069 |
| | **FP*Anecdotal** | **-3.77** | **-6.75, -0.78** | **.013** |
| Vaccination effort | Intercept (Control) | -0.54 | -2.03, 0.95 | .475 |
| | Statistical | 0.15 | -2.23, 2.53 | .899 |
| | Anecdotal | -0.67 | -2.68, 1.35 | .518 |
| | FP | -1.04 | -2.63, 0.55 | .200 |
| | FP*Statistical | 0.41 | -1.94, 2.77 | .731 |
| | **FP*Anecdotal** | **2.22** | **0.13, 4.31** | **.037** |
| Vaccine confidence | **Intercept (Control)** | **-2.28** | **-3.63, -0.93** | **.001** |
| | Statistical | 1.96 | -0.20, 4.12 | .076 |
| | Anecdotal | 0.97 | -0.86, 2.80 | .298 |
| | FP | -0.85 | -2.30, 0.59 | .247 |
| | FP*Statistical | -0.66 | -2.79, 1.48 | .548 |
| | FP*Anecdotal | 0.17 | -1.73, 2.06 | .861 |
| Disease threat | Intercept (Control) | 0.17 | -1.47, 1.80 | .841 |
| | Statistical | -0.57 | -3.19, 2.04 | .667 |
| | Anecdotal | 0.12 | -2.10, 2.34 | .918 |
| | FP | -0.41 | -2.17, 1.34 | .642 |
| | FP*Statistical | -0.49 | -3.09, 2.10 | .708 |
| | FP*Anecdotal | -1.93 | -4.22, 0.37 | .100 |
| Vaccination altruism | Intercept (Control) | 0.63 | -0.96, 2.23 | .437 |
| | **Statistical** | **4.35** | **1.80, 6.91** | **.001** |
| | Anecdotal | 1.70 | -0.47, 3.87 | .125 |
| | FP | -1.03 | -2.74, 0.68 | .236 |
| | FP*Statistical | -0.29 | -2.82, 2.24 | .820 |
| | FP*Anecdotal | -1.23 | -3.48, 1.01 | .280 |
| Frustration | **Intercept (Control)** | **2.22** | **2.00, 2.43** | **< .001** |
| | Statistical | -0.22 | -0.57, 0.13 | .218 |
| | Anecdotal | 0.22 | -0.08, 0.51 | .151 |
| | FP | -0.04 | -0.27, 0.20 | .758 |
| | **FP*Statistical** | **0.63** | **0.29, 0.97** | **< .001** |
| | **FP*Anecdotal** | **0.74** | **0.44, 1.05** | **< .001** |
| Message relevance | **Intercept (Control)** | **1.95** | **1.73, 2.17** | **< .001** |
| | **Statistical** | **2.15** | **1.79, 2.50** | **< .001** |
| | **Anecdotal** | **1.14** | **0.84, 1.45** | **< .001** |
| | FP | -0.06 | -0.30, 0.18 | .606 |
| | **FP*Statistical** | **-0.82** | **-1.17, -0.47** | **< .001** |
| | **FP*Anecdotal** | **-0.57** | **-0.88, -0.26** | **< .001** |
| Message helpfulness | **Intercept (Control)** | **1.77** | **1.55, 1.99** | **< .001** |
| | **Statistical** | **1.97** | **1.62, 2.31** | **< .001** |
| | **Anecdotal** | **0.98** | **0.69, 1.27** | **< .001** |
| | FP | -0.13 | -0.36, 0.10 | .263 |
| | **FP*Statistical** | **-0.64** | **-0.98, -0.30** | **< .001** |

(*Continued*)

**Table 8.** (Continued)

| Outcome | Predictor | b | 95% CI | p |
|---|---|---|---|---|
| | FP*Anecdotal | **-0.40** | **-0.70, -0.09** | **.010** |

Statistical = statistical group; Anecdotal = anecdotal group; Control = control group; FP = format preference; Next vaccine = intention to take the next influenza vaccine (change score); Vaccination effort = perceived effort of getting the influenza vaccine (change score); Vaccine confidence = confidence in the influenza vaccine (change score); Disease threat = perceived threat of influenza (change score); Vaccination altruism = perceived importance of getting the influenza vaccine to protect others (change score); Frustration = frustration caused by the intervention; Message relevance = perceived relevance of the intervention material; Message helpfulness = perceived helpfulness of the intervention material;

* = interaction between predictors;

bold = statistically significant effect.

analyses showed that the statistical and the anecdotal groups did not significantly differ from the control group when it comes to the change in the intention to take the next seasonal influenza vaccine, perceived effort to get an influenza vaccine, influenza vaccine confidence, or perceived influenza disease threat. However, the statistical intervention led to a greater increase in influenza vaccination altruism than the control intervention, $b = 4.35$, CI[1.80, 6.91], $p = .001$. The difference between the control group and the anecdotal group on this outcome was not statistically significant. In line with the COVID-19 experiment, the results indicated that non-tailored statistical and anecdotal interventions would not decrease vaccine hesitancy.

**Efficacy of tailored interventions.** As for the COVID-19 model, the main effects of format preference represent the associations between format preference and the outcomes in the control group. Again, none of these associations were statistically significant. The results showed that there were no statistically significant interactions between format preference and the type of intervention when it comes to the pre-post changes in influenza vaccine confidence, perceived threat of influenza, or influenza vaccination altruism. However, the relationship between format preference and the intention to take the next seasonal influenza vaccine differed between the anecdotal group and the control group, $b = -3.77$, CI[-6.75, -0.78], $p = .013$. This interaction was explained by the fact that the simple effect of format preference on the intention to take the next seasonal influenza vaccine was negative and statistically significant in the anecdotal group, $b = -3.66$, CI[-5.59, -1.74], $p < .001$, but non-significant in the control group (as stated above). The results thus unexpectedly indicated that a stronger anecdotal preference was related to a greater decrease in the intention to take the next seasonal influenza vaccine following the anecdotal intervention. The relationship between format preference and this outcome in the statistical group did not significantly differ from that in the control group. Finally, the interaction effects indicated that the relationship between format preference and the change in perceived effort to get the next seasonal influenza vaccine was significantly different in the anecdotal group compared to the control group, $b = 2.22$, CI[0.13, 4.31], $p = .037$. However, similarly to the control group, the simple effect of format preference on perceived effort to get the next seasonal influenza vaccine was not statistically significant in the anecdotal group, $b = 1.18$, CI[-0.17, 2.53], $p = .087$, suggesting no effect of format preference on this outcome among those who had received the anecdotal intervention. To sum up, these results suggested that even if the recipients were to receive influenza vaccine hesitancy interventions that match their reported format preference, the interventions would not be effective.

**Intervention reception.** The multiple regression analyses revealed that the participants neither became more nor less frustrated by the statistical intervention, $b = -0.22$, CI[-0.57, 0.13], $p = .218$, or the anecdotal intervention, $b = 0.22$, CI[-0.08, 0.51], $p = .151$, than by the

**Table 9. Self-reported influence of the interventions: The influenza experiment.**

| Response | Statistical | Anecdotal | Control |
|---|---|---|---|
| Somewhat or significantly increased intentions | 20.7% | 10.5% | 1.0% |
| Somewhat or significantly decreased intentions | 8.3% | 11.8% | 5.8% |
| Had no effect on intentions | 71.1% | 77.6% | 93.2% |

Statistical = statistical group; Anecdotal = anecdotal group; Control = control group.

control material. The anecdotal intervention was, however, perceived as more frustrating than the statistical intervention, $b = 0.43$, CI[0.10, 0.77], $p = .012$.

Both the statistical intervention and the anecdotal intervention were perceived as more relevant than the control material, $b = 2.15$, CI[1.79, 2.50], $p < .001$, and, $b = 1.14$, CI[0.84, 1.45], $p < .001$, respectively. Moreover, the anecdotal intervention was perceived as less relevant than the statistical intervention, $b = -1.00$, CI[-1.35, -0.66], $p < .001$. The participants also perceived the statistical and the anecdotal interventions as more helpful than the control material, $b = 1.97$, CI[1.62, 2.31], $p < .001$, and, $b = 0.98$, CI[0.69, 1.27], $p < .001$, respectively. The anecdotal intervention was again perceived as less helpful than the statistical intervention, $b = -0.98$, CI[-1.32, -0.65], $p < .001$.

As expected, there was no statistically significant association between format preference and frustration, message relevance or message helpfulness in the control group. However, for all those outcomes, the interactions between format preference and intervention group were statistically significant (statistical group: frustration: $b = 0.63$, CI[0.29, 0.97], $p < .001$; relevance: $b = -0.82$, CI[-1.17, -0.47], $p < .001$; helpfulness: $b = -0.64$, CI[-0.98, -0.30], $p < .001$; and the anecdotal group: frustration: $b = 0.74$, CI[0.44, 1.05], $p < .001$; relevance: $b = -0.57$, CI[-0.88, -0.26], $p < .001$; helpfulness: $b = -0.40$, CI[-0.70, -0.09], $p = .010$), suggesting that the effect of format preference on frustration, relevance and helpfulness, was different in the intervention groups compared to the controls. The simple effect of format preference on frustration was positive and statistically significant both in the statistical group, $b = 0.59$, CI[0.34–0.85], $p < .001$, and the anecdotal group, $b = 0.71$, CI[0.51, 0.90], $p < .001$, indicating that the more the participants preferred anecdotes, the more frustrated they became because of the interventions. The simple effect of format preference on perceived message relevance was also negative and statistically significant in the statistical group, $b = -0.88$, CI[-1.14, -0.62], $p < .001$, and the anecdotal group, $b = -0.63$, CI[-0.84, -0.43], $p < .001$, indicating that the more the participants preferred anecdotes, the less relevant they found the interventions to be. Finally, the simple effect of format preference on perceived message helpfulness was negative and statistically significant in the statistical group, $b = -0.77$, CI[-1.02, -0.52], $p < .001$, and in the anecdotal group, $b = -0.53$, CI[-0.72, -0.33], $p < .001$, indicating that the more participants preferred anecdotes the less helpful they found the interventions to be.

**Self-reported influence of the interventions.** The majority of the participants again stated that the interventions did not change their intentions to get vaccinated against influenza. Around twenty percent in the statistical group and 10% in the anecdotal group indicated that the interventions had increased their intentions to get vaccinated. However, approximately 10% of the participants in the statistical and the anecdotal group stated that the interventions decreased their intentions to get vaccinated against influenza (See Table 9).

## Discussion

Research suggests that vaccine-promoting anecdotal testimonies have the potential to increase the persuasiveness of interventions [19]. The persuasive benefits of anecdotal interventions compared to other kinds of interventions, however, tend to be small or very small, and the results are mixed [20]. In the present study, we investigated whether the efficacy of anecdotal and statistical vaccine hesitancy interventions is dependent on the recipients' format preference. To do this, we first developed a scale to measure preference for anecdotal and statistical information when making health decisions. The scale demonstrated good reliability and validity (see the S3 File). Then, we conducted two large-scale experiments (one experiment concerning COVID-19 vaccines and one concerning influenza vaccines) and explored whether anecdotal interventions are more successful in decreasing vaccine hesitancy in individuals who prefer anecdotal information when they make health decisions, and whether statistical interventions are more effective in those who prefer statistical information. The results did not support our hypotheses, as they showed that neither the statistical nor the anecdotal intervention decreased hesitancy toward COVID-19 and influenza vaccines, regardless of whether the participants had received interventions that were in line with their format preference or not. Instead, we found that a stronger preference for anecdotes was associated with perceiving both the statistical and the anecdotal interventions as more frustrating in the COVID-19 experiment, and less relevant and helpful in both experiments. Moreover, the more the participants preferred anecdotal information the more their intentions to take a third COVID-19 vaccine was increased by the statistical intervention and the more their intention to take the next seasonal influenza vaccine was decreased by the anecdotal intervention. It is also worth noting that approximately 25% of the participants in the COVID-19 intervention groups and 10% in the influenza intervention groups reported that the interventions decreased their intentions to get vaccinated.

The fact that the interventions were not successful in decreasing vaccine hesitancy, even when the intervention format matched the recipients' format preference, may, however, partly be due to our choice of target population. In the current study, both samples primarily consisted of participants who were very hesitant towards vaccines (66.7% of the COVID-19 sample and 62.4% of the influenza sample reported that they were less than 20% likely to take the vaccine), which is a group that is generally considered to be highly resistant to vaccine hesitancy interventions [27,28,39,40]. On the other hand, a recent randomized controlled trial, investigating the effectiveness of different pro-vaccine messages on COVID-19 vaccine hesitancy, suggests the opposite, as the messages only reduced hesitancy in participants who were highly hesitant [41]. To acquire more information on the role of the level of vaccine hesitancy, we conducted post hoc analyses where we reran the multiple regressions separately for those individuals who reported that it would be 20% or less likely that they would take the vaccines, and those who reported a likelihood of 20–79%. The pattern of the results from these two groups did not otherwise deviate from the analyses that included the whole sample, with the notable exception that influenza vaccination intentions were slightly increased by the statistical and the anecdotal interventions when compared to the control group. On the whole, the post hoc analyses suggested that the efficacy of the interventions was not related to the participants' level of vaccine hesitancy. The sample sizes in these subgroup analyses were, however, small.

Another possible explanation for the fact that none of the interventions were found effective in increasing vaccination intentions and positive vaccine attitudes can be found in the theory of attitude roots, first introduced by Hornsey and colleagues [42,43]. This theory posits that attempts to change anti-science attitudes by providing facts may turn out to be unsuccessful if the roots to these attitudes, such as fears, ideologies, worldviews, and identity, are

not addressed. It is thus possible that the interventions would have been more effective if we had aligned the interventions with potential attitude roots.

Also, due to the pre-post design of our study, vaccination intentions and vaccine attitudes were measured both before and after the intervention materials. Presenting these measures before the intervention might have affected the efficacy of the interventions by making the persuasive intent explicit. However, based on a systematic review by Graaf et al. [44] the persuasiveness of narratives in health messaging is not tied to whether these narratives are explicitly persuasive or not.

The rather unexpected finding that those who preferred anecdotes were inclined to react negatively to the anecdotal intervention, could at least partly reflect a distrust in science and the sources providing scientific information such as health authorities. This explanation is supported by the fact that a stronger self-reported anecdotal preference was strongly correlated with distrust in health authorities in the validation studies (see the S3 File). To investigate this matter, we examined whether the negative association between trust in health authorities and a preference for anecdotes could help explain why the statistical and anecdotal interventions were ineffective at increasing participants' vaccination intentions. To do this, we conducted post hoc regression analyses, in which we added trust in health authorities as a control variable to the COVID-19 and influenza vaccination intention models. These analyses showed that trust in health authorities was directly associated with vaccination intentions. Trust in health authorities did, however, not noteworthily change the relationships between format preference, intervention group, and vaccination intentions, nor did it change the statistical significance levels of these relationships. In other words, the inefficacy of the format-preference-tailored vaccine hesitancy interventions did not seem to be tied to the participants' level of trust in health authorities.

Format preference may, however, be related to reactance (i.e., negative emotions that may arise when people feel that their freedom of choice has been threatened [26]), as those who preferred anecdotes often had negative reactions towards the statistical and the anecdotal intervention materials (i.e., attempts to influence participants' vaccine attitudes), but not towards the control material (i.e., no attempt to influence the participants). Despite the negative reactions towards the interventions, the results revealed no decrease in the participants' vaccine attitudes and vaccination intentions from pretest to posttest, as has been seen in some other studies [27,28]. It is, however, important to note that, already before the intervention, most participants considered it very unlikely that they would get vaccinated. This floor effect might conceal a possible backfire effect. Further indication of this is that approximately 25% of the participants in the COVID-19 interventions stated that the interventions decreased their willingness to get vaccinated. To investigate this further, we conducted post hoc t-tests in which we compared the pre-intervention vaccination intentions between participants who had reported that the interventions had decreased their vaccination intentions and those who had reported that the interventions had either increased or had not affected their intentions. The results showed that the participants who reported that the interventions had decreased their vaccination intentions were also less likely to take a seasonal COVID-19 vaccine, a third COVID-19 vaccine, and the next seasonal influenza vaccine prior to the interventions. This finding supports our floor effect suspicions and is in line with the previous studies, suggesting that vaccine hesitancy interventions might backfire when presented to highly vaccine hesitant people [27,28].

It is worrying that the participants responded negatively to both the statistical and anecdotal interventions, particularly in the COVID-19 interventions. However, the statistical interventions were consistently considered less frustrating, more relevant, and more helpful than the anecdotal interventions in both samples. This suggests that even if neither type of intervention decreased vaccine hesitancy, a statistical intervention might be slightly less prone to backfire

and may therefore be a better option when designing purely statistical or anecdotal vaccine hesitancy interventions. It is further worth noting that the employed intervention materials were most likely familiar to the participants prior to the intervention, as vaccine messaging and general discussion on vaccines, especially about the COVID-19 vaccines, has been unprecedentedly rampant due to the COVID-19 pandemic. It is thus likely that most of our participants had come into contact with COVID-19 vaccine statistics and COVID-19 vaccination anecdotes before participating in this study, which could have affected the efficacy of the interventions. The fact that significantly more participants in the COVID-19 group, compared to the influenza group, reported that the interventions had at least somewhat decreased their intentions could be a sign of message fatigue.

Finally, during the scale validation process, we found that negative vaccine anecdotes were more persuasive than positive ones in affecting vaccine attitudes against made-up diseases. This points to a negativity bias, which refers to the tendency to react more strongly to negative than positive information [45]. This finding is in line with previous research, as similar results have been reported in other studies [13,46–48]. Thus, our results imply that even if anti-vaccination anecdotes do increase vaccine hesitancy, the use of anecdotal pro-vaccination messages may not be as efficient in decreasing it.

## Limitations

This study has some limitations that are worth noting. First, this study used a convenience sample, consisting of social media users in Finland. However, as we state above, Finnish Facebook users alone account for a large proportion of the Finnish population. Moreover, since we excluded participants who were likely to take a seasonal COVID-19 vaccine or the next seasonal influenza vaccine, we do not know how people with high vaccination intentions would have responded to the interventions. Next, though the anecdotes in the anecdotal interventions were carefully crafted from real vaccine anecdotes found on the Internet, and were created to be as authentic as possible, it is possible that they were not appealing or believable enough to influence the respondents' vaccine attitudes and vaccination intentions. Furthermore, while previous studies have shown that combinations of statistical and anecdotal interventions can potentially be more effective than purely statistical and anecdotal interventions [20], we only studied these interventions separately. Further, the participants did not choose the format of the intervention they received. Instead, we examined how people with different format preferences responded to statistical and anecdotal interventions. Also, the fact that the anecdotal and control conditions employed the same pictures, while different pictures were used for the statistical interventions, could have affected the results. However, using the same photos in the anecdotal and control conditions excludes the possibility that any effect of the anecdotal intervention compared to the control intervention, would be due to the photos used. As statistics are not linked to specific people, we used thematic pictures in the statistical intervention to keep the anecdotal and statistical interventions as similar as possible. Moreover, the vaccine attitude and vaccination intention measures were based on self-reports, making them susceptible to biases, such as the desirability bias. Lastly, we measured only vaccine attitudes and vaccination intentions instead of actual vaccine uptake. Thus, we encourage future studies to longitudinally investigate how vaccine hesitancy interventions, that have been tailored according to people's format preference, affect actual vaccine uptake.

## Conclusions

In the present study, the statistical and anecdotal vaccine hesitancy interventions turned out to be ineffective on their own at decreasing COVID-19 and influenza vaccine hesitancy, even

when people received interventions in their preferred format. This finding suggests that tailoring vaccine hesitancy interventions according to format preference—as measured in the present study—might not increase the efficacy of the interventions, particularly in individuals who are highly hesitant. In fact, people who preferred anecdotes over statistics tended to react negatively toward both interventions, but even more so toward the anecdotal intervention. Taken together, this study highlights a need for caution when using anecdotal testimonies to persuade people who hold anti-vaccination attitudes and suggests that purely statistical vaccine hesitancy interventions might have a small edge over purely anecdotal interventions, due to the statistical interventions being slightly less likely to backfire.

## Supporting information

**S1 File. Power analysis.**
(PDF)

**S2 File. Supporting information for the COVID-19 and influenza experiments.**
(DOCX)

**S3 File. Supporting information for the format preference scale validation studies.**
(DOCX)

## Author Contributions

**Conceptualization:** Karl O. Mäki, Linda C. Karlsson, Johanna K. Kaakinen, Philipp Schmid, Stephan Lewandowsky, Jan Antfolk, Anna Soveri.

**Formal analysis:** Karl O. Mäki, Linda C. Karlsson, Johanna K. Kaakinen, Jan Antfolk.

**Methodology:** Karl O. Mäki, Philipp Schmid, Stephan Lewandowsky, Jan Antfolk, Anna Soveri.

**Project administration:** Anna Soveri.

**Resources:** Anna Soveri.

**Supervision:** Linda C. Karlsson, Johanna K. Kaakinen, Anna Soveri.

**Visualization:** Karl O. Mäki.

**Writing – original draft:** Karl O. Mäki, Anna Soveri.

**Writing – review & editing:** Karl O. Mäki, Linda C. Karlsson, Johanna K. Kaakinen, Philipp Schmid, Stephan Lewandowsky, Jan Antfolk, Anna Soveri.

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
