## [Decision Letter · Decision Letter 0]

12 Oct 2022

PONE-D-22-18819Tailoring Interventions to Suit Self-Reported Format Preference Does Not Decrease Vaccine HesitancyPLOS ONE

Dear Dr. Mäki,

Thank you for submitting your manuscript to PLOS ONE. After careful consideration, we feel that it has merit but does not fully meet PLOS ONE’s publication criteria as it currently stands. Therefore, we invite you to submit a revised version of the manuscript that addresses the points raised during the review process.Please ensure that your decision is justified on PLOS ONE’s publication criteria and not, for example, on novelty or perceived impact.

We look forward to receiving your revised manuscript.

Kind regards,

Jonathan Jong, PhD

Academic Editor

PLOS ONE

Journal Requirements:

“This project has received funding from the European Union’s Horizon 2020 research and innovation program under grant agreement No 964728 (JITSUVAX). KOM was funded by the Faculty of Social Sciences at the University of Turku (www.utu.fi/en/university/faculty-of-social-sciences). JKK and KOM received funding from the Strategic Research Council’s LITERACY program (Academy of Finland grant number: 335233). AS was funded by the Academy of Finland (grant number: 316004; www.aka.fi/en/). SL also acknowledges funding from the Humboldt Foundation Germany through a research award.

NO”

4. Please upload a new copy of Figure 1 as the detail is not clear. Please follow the link for more information: " ext-link-type="uri" xlink:type="simple">https://blogs.plos.org/plos/2019/06/looking-good-tips-for-creating-your-plos-figures-graphics/"
https://blogs.plos.org/plos/2019/06/looking-good-tips-for-creating-your-plos-figures-graphics/

Reviewers' comments:

Reviewer's Responses to Questions

**Comments to the Author**

1. Is the manuscript technically sound, and do the data support the conclusions?

Reviewer #1: Partly

Reviewer #2: Yes

Reviewer #3: Yes

2. Has the statistical analysis been performed appropriately and rigorously? 

Reviewer #1: Yes

Reviewer #2: Yes

Reviewer #3: Yes

3. Have the authors made all data underlying the findings in their manuscript fully available?

Reviewer #1: Yes

Reviewer #2: Yes

Reviewer #3: Yes

4. Is the manuscript presented in an intelligible fashion and written in standard English?

Reviewer #1: Yes

Reviewer #2: Yes

Reviewer #3: Yes

5. Review Comments to the Author

Reviewer #1: Overall, I found the paper well-written and based on a theoretically and practically interesting idea. The author did a good job explaining the main idea in the introduction, clearly describing the experimental and statistical procedure, and including two sets of experimental stimuli (COVID vaccines and influenza vaccines) for replication. Despite many things to like about this paper, I do have several major concerns and suggestions for possible improvement.

I feel the literature review could be much more expanded to properly theorize the concepts of interest and predicted effects. Specifically, although the author did a great job reviewing the most recent and relevant empirical evidence regarding statistical vs. anecdotal messages and tailored intervention, the psychological mechanisms underlying their respective persuasive advantages could be briefly explained so that readers have adequate knowledge of why their effects are worth testing. Moreover, some of the outcome variables (except most obvious ones such as attitudes and intentions) should be properly defined and their connection to narrative and/or tailored persuasion can be explained. This would justify why they were studied in the research. And it could perhaps be done as the author explains the mechanisms of narrative and tailored effects. For example, narratives are considered to induce less audience reactance (frustration) and tailored messages are considered to be more relevant. Relatedly, a list of hypotheses should be developed based on reviewed literature and logical reasoning. This helps readers clearly understand the aims of the statistical tests.

I am concerned about the measures used in the study. Most of the measures were not from established scales (Although I applauded the effort to develop and validate the format preference scale). Some of the measures used single items (e.g., relevance, helpfulness, frustration, intention); and some apparently lack face validity. For example, the vaccine attitudes measure included perceived threat of the disease and perceived ease of vaccination effort. Although the author did a CFA to make sure the statistical loadings made sense and removed the effort item, I don’t see how the threat item contribute to the operationalization of vaccine attitudes in a theoretical sense. Perceived threat is a conceptually distinct variable.

The research yielded many null findings. As the author did not formally theorize and propose a list of hypotheses, it is relatively difficult to judge if these findings were aligned with the expectations and what their theoretical implications were. I’m okay with null findings but they would be better grounded in theories. Otherwise, I would have questions about the actual contribution of the study to knowledge.

In my opinion, the scale development makes a better contribution to the literature on message format than the null findings actually reported in the paper. I wonder if the manuscript could be restructured to report both. I do see some of the findings in the scale development section help explain some of the null findings. For example, preference for anecdotal messages was negatively related to some of the outcome variables directly. The third variable here might be trust in medicine or government. This variable might be negatively related to anecdotal preference and positively related to vaccination outcomes. Thus, the author observed the reported negative association. I also think that trust should be controlled in analyses if it was measured in the experiments to rule out the possibility. Correspondingly, if the scale development is included as a first study, a better job needs to be done to conceptualize format preference in the literature review. Of course, I understand whether or not to do this depends on the length limit. But I think this presents a better contribution and could be done by reducing the descriptions of the statistical procedures. Those parts may be better included in supplemental materials or footnotes to save space for important findings and theory development.

I also listed my other thoughts and suggestions:

Please provide justification for using a convenient sample recruited on social media.

Please explain why 80% of likelihood to get vaccinated was used as a cut-off point.

Please discuss how measuring vaccine attitudes prior to message exposure influence narrative persuasion (known to be more powerful when persuasive intent is implicit) and potential testing effects.

Any control variables included? For example, prior vaccination status, trust in science/medicine/government, etc.

Please explain why photos were not used in the statistical intervention but in the control material and its possible confounding effect.

The message details in Figure 1 were not readable. Please provide the full messages in supplemental materials along with English translations.

Please include the mean tables and descriptive statistics of the variables (reliability, mean, SD) in the main text.

Please revisit the discussion after putting together a list of hypotheses and discuss the theoretical implications.

Please acknowledge limitations related to the sample and measures.

Reviewer #2: Timely topic to examineas cuntries explore how best to catch up routime immunization as wellas immunize adults with flu and COVID vaccines.

A concern from the begining - vaccine decsion making has many inlfuences - ie many factors inlfuence - this study examines one set of factors stats vs anecdote - imp point to understand BUT in real world there would be context, what others in social network doing, personal past expereinces , etcetc that can inlfuence that decsion. Not sure is possible to keep all of these bkg factors similar across a study.However- helpful to know how this one variable - stats vs story impacted

in Methods - line 148-9 recruitment - do not know bkg in Finland to know access and use of these platforsm by elderly for example

- line 177 - how did you measure frustration? Is it a validated tecnique?

- Figure 1 - these were unreadable even when a snipped them and increased the size ....are they in Finish? Figure needs ot redone in font sie that is readable - a key element in the sytudy is the intervention and that is not clear for review as presented

- line 200-220- pre post measures - has been controversy in literatire whether "intent" to accpe timmunization is strngly correalted with action . So imp here to note this, justfy why chose intent and also that prepost survey questions validated . The next section dicussed reliability BUT that is to intent not in actual uptake - do you know if intent measures her are linked to actual uptakr ?

Lines 229 -231- this is not a surprise as negative st=ciks more than positive and framing matters - why is so key to be able to actually read the anecdotes to see how frmaed and whther +ve or negative

lines 236 -242...again were these validated ?

Lines 264-5. " strongly skewed and even slightly bimodal at times" one might have exepcted this esp if negative anexdotes vs positive or neutral ie stats given the stikiness of negative stories - helpful to see adjusted for this

Results

305-308....for me this is not a great surproise- why wanted to know if had validated before actual study

302-319. this is so imp and strengthens the study results

326-328 - since vaccine decsion making is influenced by many factors - perhaps this is not such a surprise

line 359-364...there was no both interventions together vs control - that would have been interesting ....not clear in methods why this was not another arm of the study.....esp as you found that more participants preferred anecdotes, the more their intentions to take a third COVID-19 vaccine increased by the statistical intervention. Does this notagree with marketing evidence that data tells but stories sell and if put together even more powerful? Why an arm that had both would have added value

Lines 396 to 400 - in this 25 % negative impact gorup - were these negative data and/or anecdotes ie the stickiness issue ?

Flu - very helpful to see the flu data paralelled COVID data in some waysbut differed in other. This may well reflect different perceived context and hence differing importance between flua nd COVID vaccines, diff past experiences etcetc ie th emyriad other factors know impacts vaccien decsion making . This strengthen the paper

Discussion 527-530. Most studies of single interventions not change vaccine acceptance much - so not a big surprise here . Do wish had had 4th arm of stats + anecdote vs control

576 -579 - yes context and expereince do matter ....

591-595- as I could not read the anecdotes - don't know but likley correct

607 -608 might want to be more nuanced than this - stats or anecdotes indodulally did not appear to be effective in pop where not very postive torwards these vaccines.

612-615- not sure can go that far- as never tested the two together - that has to eb one of the conlusions that that nees to be done

Reviewer #3: Review for PONE-D-22-18819

This paper has two aims. First it develops a scale that aims to measure self-reported preferences for statistical or anecdotal information amongst vaccine hesitant individuals. Second it tests the interaction between format preferences (statistical or anecdotal information) and format type (statistical or anecdotal) on vaccine attitudes and intentions amongst vaccine deniers. The authors predicted that congruent information preference and intervention format (prefer statistical--informed statistical vs. preferred anecdotal--informed anecdotal) would boost vaccine intentions amongst vaccine hesitant participants. They found no support for their predictions. A non-predicted finding was that participants with preferences for anecdotes were more frustrated by the interventions regardless of format type (statistical, anecdotal) and found the interventions less helpful and relevant. Furthermore, in contrast to the predictions, the authors found that statistical information increased vaccine intentions of participants with higher preferences for anecdotal information. The topic of the paper is timely.

Theoretical background

It is unclear what the theoretical support for the predictions is. Why did the authors theoretically predict that interventions will be more effective in promoting vaccine intentions when the intervention format was congruent with the participants' format preference? And specifically, why did the authors predict this for vaccine denier? The paper would benefit from a clearer theoretical account. Without this it seems exploratory which also would be fine but then this needs to be clearly stated.

Methods

The preregistration OSF link presented in the paper did not work (I only found a OSF link for the data). For future submissions, I recommend that the authors are careful about including a link in their method section that works. It is crucial in judging the work. As a reviewer I need to understand whether the predictions made in the paper were a-priori or post-hoc, what inclusion and exclusion criteria were preregistered, whether the analyses were planned and what variables were used. I was unable to judge this as the link did not work.

[Page 8, line 162 to 165] The power simulation presented in the paper seems undetailed. It would be helpful for the reader to include more details about what estimates were used and where they were based on.

[Page 8, line 165 to 167] The exclusions of 843 participants is a very large part of the sample (43.4%). It would be important to understand whether this was preregistered and anticipated. But because the preregistration link did not work I am unable to judge this.

Materials

It is great that the authors included examples of the materials that were used, but they are not readable in the current format. It would be important to include them in a higher quality so people can read and judge them. This is especially important given the null effect of the present study, which is informative for people who might want to test a similar intervention.

Discussion

The authors might wish to be more careful about their conclusions. They could for example be more careful in their wording and include the following suggestion or something along those lines (see **WORDING BELOW**):

[Page 30, line 526 to 528] “The present experiment consistently suggests that it is unlikely that tailoring vaccine hesitancy interventions according to format preferences **as measured in the present study with the present sample**,would enhance the efficacy of the interventions.

I am wondering whether the intervention would be effective for uncertain participants. This group might benefit from tailored interventions. Is there a theoretical account that would suggest this?

Best of luck with this research!

6. PLOS authors have the option to publish the peer review history of their article (what does this mean?). If published, this will include your full peer review and any attached files.

Reviewer #1: No

Reviewer #2: No

Reviewer #3: No

---

## [Author Response · Author response to Decision Letter 0]

21 Nov 2022

Response to Reviewers

General: We thank the Reviewers for their constructive and valuable comments. As a general note, we have split the Supplementary into three files named S1 File, S2 File, and S3 File, to make tables and figures easier to find and to match PLOS ONE’s supplementary referencing guidelines. Please note that the line numbers in our responses match the Revised Manuscript with tracked changes on.

Reviewer #1

Comment 1: Overall, I found the paper well-written and based on a theoretically and practically interesting idea. The author did a good job explaining the main idea in the introduction, clearly describing the experimental and statistical procedure, and including two sets of experimental stimuli (COVID vaccines and influenza vaccines) for replication. Despite many things to like about this paper, I do have several major concerns and suggestions for possible improvement.

I feel the literature review could be much more expanded to properly theorize the concepts of interest and predicted effects. Specifically, although the author did a great job reviewing the most recent and relevant empirical evidence regarding statistical vs. anecdotal messages and tailored intervention, the psychological mechanisms underlying their respective persuasive advantages could be briefly explained so that readers have adequate knowledge of why their effects are worth testing. Moreover, some of the outcome variables (except most obvious ones such as attitudes and intentions) should be properly defined and their connection to narrative and/or tailored persuasion can be explained. This would justify why they were studied in the research. And it could perhaps be done as the author explains the mechanisms of narrative and tailored effects. For example, narratives are considered to induce less audience reactance (frustration) and tailored messages are considered to be more relevant. Relatedly, a list of hypotheses should be developed based on reviewed literature and logical reasoning. This helps readers clearly understand the aims of the statistical tests.

Response: Based on the Reviewer’s suggestion, we have added three segments to the Introduction in which we describe variables and current knowledge surrounding the underlying mechanisms of the effects of statistical, anecdotal, and tailored interventions. The first segment can be found on lines 74–79: “Whereas statistical information is theorized to be persuasive because people perceive it as credible, verifiable, and generalizable [16], the persuasiveness of narratives has been attributed to their ability to reduce psychological reactance and counterarguing and to enable engagement in a storyline and identification with characters [17], while simultaneously being more easily processed than other types of formats [18].”. The second segment can be found on the lines 108–112: “Designing tailored interventions is believed to increase recipients’ attention to the message and by extension to make the message easier to remember. Tailored interventions are also expected to more often be perceived as relevant and to elicit more thorough elaboration of tailored messages compared to non-tailored ones [24].”, and the third segment can be found on the lines 131–134: “ We furthermore investigated whether recipients’ format preference was related to how the messages were received by the participants, that is, to what degree they experienced the information in the interventions to be relevant, helpful, and/or frustrating.”. Finally, we added a list of hypotheses to the end of the Introduction, lines 144–157.

Comment 2: I am concerned about the measures used in the study. Most of the measures were not from established scales (Although I applauded the effort to develop and validate the format preference scale). Some of the measures used single items (e.g., relevance, helpfulness, frustration, intention); and some apparently lack face validity. For example, the vaccine attitudes measure included perceived threat of the disease and perceived ease of vaccination effort. Although the author did a CFA to make sure the statistical loadings made sense and removed the effort item, I don’t see how the threat item contribute to the operationalization of vaccine attitudes in a theoretical sense. Perceived threat is a conceptually distinct variable.

Response: We thank the Reviewer for recognizing the effort put into the format scale development. We also agree with the Reviewer that single-item measures can be problematic. When it comes to vaccination intention, message relevance, helpfulness, and frustration, we, however, concluded that single-item measures were sufficient to get a general understanding of the participants’ vaccination intentions and how they perceived the intervention. This decision was made to keep our experiment as short as possible, as too long questionnaires would result in larger drop-out rates and respondent fatigue. The vaccine attitude items, on the other hand, were preregistered to be used to create a factor, but as presented in the manuscript (lines 377–384, and 511–517), the fit indices and factor loadings did not support the factor solutions (possibly for the reasons pointed out by the Reviewer). Because of that we decided to treat them as separate constructs. 

Comment 3: The research yielded many null findings. As the author did not formally theorize and propose a list of hypotheses, it is relatively difficult to judge if these findings were aligned with the expectations and what their theoretical implications were. I’m okay with null findings but they would be better grounded in theories. Otherwise, I would have questions about the actual contribution of the study to knowledge.

Response: We apologize for failing to have made the preregistration document available prior to the review process. It can now be found behind the link in the manuscript (osf.io/jqspe). Our hypotheses were preregistered, and we have tried to clarify them in the Introduction (lines 125–129; 136–137; 141–142), and we added a comprised list of the hypotheses to the end of the introduction, lines 144–157. Furthermore, we have added to the Discussion section that the results did not match our hypotheses, lines 650–651: “The results did not support our hypotheses, as they showed…”.

Comment 4: In my opinion, the scale development makes a better contribution to the literature on message format than the null findings actually reported in the paper. I wonder if the manuscript could be restructured to report both. I do see some of the findings in the scale development section help explain some of the null findings. For example, preference for anecdotal messages was negatively related to some of the outcome variables directly. The third variable here might be trust in medicine or government. This variable might be negatively related to anecdotal preference and positively related to vaccination outcomes. Thus, the author observed the reported negative association. I also think that trust should be controlled in analyses if it was measured in the experiments to rule out the possibility. Correspondingly, if the scale development is included as a first study, a better job needs to be done to conceptualize format preference in the literature review. Of course, I understand whether or not to do this depends on the length limit. But I think this presents a better contribution and could be done by reducing the descriptions of the statistical procedures. Those parts may be better included in supplemental materials or footnotes to save space for important findings and theory development.

Response: We have been struggling to find the best structure for the manuscript. While writing the manuscript, we realized that including the three validation studies in the main text would make the manuscript too heavy for the reader. Because of that, we made the decision to move the validation studies to the supplementary. In our opinion, this decision made the manuscript more straightforward and easier to read. Although we appreciate the recognition of our format preference scale, we also think that the null findings of the main studies are an important contribution to the literature, as there are no previous studies that combine message tailoring with statistical and anecdotal message formats in a vaccine hesitancy context. With this in mind, we have chosen not to include the validation studies in the main text in the revision. However, if the Reviewer still thinks the best solution is to move the validation studies from the supplementary, we will change the manuscript accordingly. We agree with the Reviewer that trust in health authorities could have had an effect on the outcomes. As the Reviewer suggested, we added trust in health authorities as a control variable to the vaccination intentions models, to examine whether it might be masking some relationships between the other variables. Since we did not preregister trust in health authorities as a control variable, we ran these analyses as post hoc analyses and report them as such in the Discussion section, lines 704–721: “The rather unexpected finding that those who preferred anecdotes were inclined to react negatively to the anecdotal intervention, could at least partly reflect a distrust in science and the sources providing scientific information such as health authorities. This explanation is supported by the fact that a stronger self-reported anecdotal preference was strongly correlated with distrust in health authorities in the validation studies (see the S3 File). To investigate this matter, we examined whether the negative association between trust in health authorities and a preference for anecdotes could help explain why the statistical and anecdotal interventions were ineffective at increasing participants’ vaccination intentions. To do this, we conducted post hoc regression analyses, in which we added trust in health authorities as a control variable to the COVID-19 and influenza vaccination intention models. These analyses showed that trust in health authorities was directly associated with vaccination intentions. Trust in health authorities did, however, not noteworthily change the relationships between format preference, intervention group, and vaccination intentions, nor did it change the statistical significance levels of these relationships. In other words, the inefficacy of the format-preference-tailored vaccine hesitancy interventions did not seem to be tied to the participants’ level of trust in health authorities. 

 Format preference may, however, be related to reactance…”.

Comment 5: I also listed my other thoughts and suggestions:

Please provide justification for using a convenient sample recruited on social media.

Response: We have added a segment to the Method section describing our choice of data collection method, lines 199–202: “This was considered an efficient and cost-effective way to collect data from the Finnish population. In 2021, Finnish Facebook users were estimated to account for approximately 63.5% of the total population, with age groups between 18 and 65+ being well represented [36].”.

Comment 6: Please explain why 80% of likelihood to get vaccinated was used as a cut-off point.

Response: The preregistration document is now available, in which the cut-off point is mentioned. Further, we have now tried to make the rationale behind this cut-off point clearer in the manuscript. Method section, lines 210–213: “To decrease the risk of ceiling-effects and to avoid unnecessary interventions, we determined that an appropriate inclusion criterion was that participants would have to be less than 80% likely to get vaccinated. This decision was made prior to the data collection.”.

Comment 7: Please discuss how measuring vaccine attitudes prior to message exposure influence narrative persuasion (known to be more powerful when persuasive intent is implicit) and potential testing effects.

Response: We added a segment to the Discussion in which we discuss the explicitly persuasive nature of our design. Discussion section, lines 697–702: “Also, due to the pre-post design of our study, vaccination intentions and vaccine attitudes were measured both before and after the intervention materials. Presenting these measures before the intervention might have affected the efficacy of the interventions by making the persuasive intent explicit. However, based on a systematic review by Graaf et al. [44] the persuasiveness of narratives in health messaging is not tied to whether these narratives are explicitly persuasive or not.”.

Comment 8: Any control variables included? For example, prior vaccination status, trust in science/medicine/government, etc.

Response: Please see our response to Reviewer #1 Comment 4.

Comment 9: Please explain why photos were not used in the statistical intervention but in the control material and its possible confounding effect.

Response: It is true that the differences in the images used in the anecdotal and control condition vs the statistical condition, may have created a confound in the study. The reason for using photos of individuals in the anecdotal and control conditions, is that we wanted to add to the realism and the ecological validity of the study. We now mention this in the Limitations section, lines 783–789: “Also, the fact that the anecdotal and control conditions employed the same pictures, while different pictures were used for the statistical interventions, could have affected the results. However, using the same photos in the anecdotal and control conditions excludes the possibility that any effect of the anecdotal intervention compared to the control intervention, would be due to the photos used. As statistics are not linked to specific people, we used thematic pictures in the statistical intervention to keep the anecdotal and statistical interventions as similar as possible.”. 

Comment 10: The message details in Figure 1 were not readable. Please provide the full messages in supplemental materials along with English translations.

Response: All intervention materials (high picture quality) and corresponding English translations have been added to the S2 File. Additionally, Fig 1 has been updated to a readable, English translation, high-quality picture of the COVID-19 intervention materials.

Comment 11: Please include the mean tables and descriptive statistics of the variables (reliability, mean, SD) in the main text.

Response: Based on the Reviewer’s suggestion, reliability and mean tables have been moved from the supplemental materials to the Results section in the main text (lines 396, 416, 527, and 542). 

Comment 12: Please revisit the discussion after putting together a list of hypotheses and discuss the theoretical implications.

Response: As the Reviewer suggested, we have now connected our findings more clearly to current theories and previous research findings, Discussion section, lines 650–651: “The results did not support our hypotheses, as they showed…”; lines 676–679: “On the other hand, a recent randomized controlled trial, investigating the effectiveness of different pro-vaccine messages on COVID-19 vaccine hesitancy, suggests the opposite, as the messages only reduced hesitancy in participants who were highly hesitant [41].”; and lines 689–702: “Another possible explanation for the fact that none of the interventions were found effective in increasing vaccination intentions and positive vaccine attitudes can be found in the theory of attitude roots, first introduced by Hornsey and colleagues [42,43]. This theory posits that attempts to change anti-science attitudes by providing facts may turn out to be unsuccessful if the roots to these attitudes, such as fears, ideologies, worldviews, and identity, are not addressed. It is thus possible that the interventions would have been more effective if we had aligned the interventions with potential attitude roots. Also, due to the pre-post design of our study, vaccination intentions and vaccine attitudes were measured both before and after the intervention materials. Presenting these measures before the intervention might have affected the efficacy of the interventions by making the persuasive intent explicit. However, based on a systematic review by Graaf et al. [44] the persuasiveness of narratives in health messaging is not tied to whether these narratives are explicitly persuasive or not.”.

Comment 13: Please acknowledge limitations related to the sample and measures.

Response: Sample and measure limitation are now acknowledged in the Limitations section, lines 769–774: “First, this study used a convenience sample, consisting of social media users in Finland. However, as we state above, Finnish Facebook users alone account for a large proportion of the Finnish population. Moreover, since we excluded participants who were likely to take a seasonal COVID-19 vaccine or the next seasonal influenza vaccine, we do not know how people with high vaccination intentions would have responded to the interventions.”, lines 790–791: “Moreover, the vaccine attitude and vaccination intention measures were based on self-reports, making them susceptible to biases, such as the desirability bias.”, and lines 796–797: “Lastly, we measured only vaccine attitudes and vaccination intentions instead of actual vaccine uptake.”. 

Reviewer #2

Comment 1: Timely topic to examineas cuntries explore how best to catch up routime immunization as wellas immunize adults with flu and COVID vaccines.

A concern from the begining - vaccine decsion making has many inlfuences - ie many factors inlfuence - this study examines one set of factors stats vs anecdote - imp point to understand BUT in real world there would be context, what others in social network doing, personal past expereinces , etcetc that can inlfuence that decsion. Not sure is possible to keep all of these bkg factors similar across a study.However- helpful to know how this one variable - stats vs story impacted

Response: To highlight the complexity of vaccine hesitancy, we added a segment to the beginning of the Introduction, lines 57–60: “Vaccine hesitancy is often used as an umbrella term for all types of attitudes and behaviors that question or go against vaccines [1]. It is a complex phenomenon that varies across different contexts and vaccines [2].”. Furthermore, we have added a segment to the Discussion section in which we discuss the fact that we only focus on some aspects of the complex phenomenon, lines 689–696: “Another possible explanation for the fact that none of the interventions were found effective in increasing vaccination intentions and positive vaccine attitudes can be found in the theory of attitude roots, first introduced by Hornsey and colleagues [42,43]. This theory posits that attempts to change anti-science attitudes by providing facts may turn out to be unsuccessful if the roots to these attitudes, such as fears, ideologies, worldviews, and identity, are not addressed. It is thus possible that the interventions would have been more effective if we had aligned the interventions with potential attitude roots.”. 

Comment 2: in Methods - line 148-9 recruitment - do not know bkg in Finland to know access and use of these platforsm by elderly for example

Response: We have added a description of Finnish Facebook users to the Method section, lines 200–202: “In 2021, Finnish Facebook users were estimated to account for approximately 63.5% of the total population, with age groups between 18 and 65+ being well represented [36].”.

Comment 3: - line 177 - how did you measure frustration? Is it a validated tecnique?

Response: This was not a validated measure, but the items in question were created specifically for this study. This is now stated in the manuscript, line 305–306: “Three Likert scale measures were created and used in this study to assess the participants’ reactions to the intervention materials”.

Comment 4: - Figure 1 - these were unreadable even when a snipped them and increased the size ....are they in Finish? Figure needs ot redone in font sie that is readable - a key element in the sytudy is the intervention and that is not clear for review as presented

Response: We wish to apologize for the unreadable figure. We have added new English translation, high-quality pictures of the COVID-19 intervention materials to the manuscript as well as high-quality pictures of all intervention materials with English translations to the S2 File.

Comment 5: - line 200-220- pre post measures - has been controversy in literatire whether "intent" to accpe timmunization is strngly correalted with action . So imp here to note this, justfy why chose intent and also that prepost survey questions validated . The next section dicussed reliability BUT that is to intent not in actual uptake - do you know if intent measures her are linked to actual uptakr ?

Response: We have clarified this limitation in the Limitations section and added a suggestion for future studies to investigate how these interventions affect actual vaccine uptake, lines 796–800: “Lastly, we measured only vaccine attitudes and vaccination intentions instead of actual vaccine uptake. Thus, we encourage future studies to longitudinally investigate how vaccine hesitancy interventions, that have been tailored according to people’s format preference, affect actual vaccine uptake.”.

Comment 6: Lines 229 -231- this is not a surprise as negative st=ciks more than positive and framing matters - why is so key to be able to actually read the anecdotes to see how frmaed and whther +ve or negative

Response: While the anecdotes were framed as either positive or negative toward vaccines in the validation studies described in the S3 File, only vaccine positive anecdotes were presented in the actual experiment.

Comment 7: lines 236 -242...again were these validated ?

Response: See previous response to Reviewer #2, Comment 3.

Comment 8: Lines 264-5. " strongly skewed and even slightly bimodal at times" one might have exepcted this esp if negative anexdotes vs positive or neutral ie stats given the stikiness of negative stories - helpful to see adjusted for this

Response: We did not present negative anecdotes in the main experiments. It seemed that participants were either strongly intent on taking the vaccines or they were strongly against it, thus resulting in a slightly bimodal distribution, even when slicing the top 80%. Hopefully this information clarifies the results.

Comment 9: Results

305-308....for me this is not a great surproise- why wanted to know if had validated before actual study

Response: We have added to the Method section that the pre- and posttest measures were created for this study. Lines 266–267: “These measures were created for this study.”. Please also see our response to Reviewer #1, Comment 2.

Comment 10: 302-319. this is so imp and strengthens the study results

Comment 11: 326-328 - since vaccine decsion making is influenced by many factors - perhaps this is not such a surprise

Response: We tried to consider the complexity of vaccine hesitancy by utilizing a sledgehammer approach in the interventions by including information about vaccine safety, vaccine efficacy, disease threat, and collective responsibility in both the statistical and anecdotal intervention materials. We did, however, not take possible attitude roots (Hornsey Fielding, 2017; Hornsey, 2020) into account, and this is now mentioned in the Discussion section as a possible explanation for null results, lines 689–696: “Another possible explanation for the fact that none of the interventions were found effective in increasing vaccination intentions and positive vaccine attitudes can be found in the theory of attitude roots, first introduced by Hornsey and colleagues [42,43]. This theory posits that attempts to change anti-science attitudes by providing facts may turn out to be unsuccessful if the roots to these attitudes, such as fears, ideologies, worldviews, and identity, are not addressed. It is thus possible that the interventions would have been more effective if we had aligned the interventions with potential attitude roots.”.

Comment 12: line 359-364...there was no both interventions together vs control - that would have been interesting ....not clear in methods why this was not another arm of the study.....esp as you found that more participants preferred anecdotes, the more their intentions to take a third COVID-19 vaccine increased by the statistical intervention. Does this notagree with marketing evidence that data tells but stories sell and if put together even more powerful? Why an arm that had both would have added value

Response: As the Reviewer suggests, having had an aggregate group where participants receive both interventions would have been very interesting. However, doing so would have required larger samples and would have been outside the scope of this study, in which we investigated format preference -tailored interventions. We have added this to the Limitations section, lines 778–781: “Furthermore, while previous studies have shown that combinations of statistical and anecdotal interventions can potentially be more effective than purely statistical and anecdotal interventions [20], we only studied these interventions separately.”.

Comment 13: Lines 396 to 400 - in this 25 % negative impact gorup - were these negative data and/or anecdotes ie the stickiness issue ?

Response: To shed more light on this finding, we have conducted post hoc t-tests and added a segment to the Discussion section in which we discuss the relationship between the self-reported impact of interventions and the pre-intervention intention to receive vaccines. We hope that this answers the Reviewers question. Discussion section, lines 733–742: “To investigate this further, we conducted post hoc t-tests in which we compared the pre-intervention vaccination intentions between participants who had reported that the interventions had decreased their vaccination intentions and those who had reported that the interventions had either increased or had not affected their intentions. The results showed that the participants who reported that the interventions had decreased their vaccination intentions were also less likely to take a seasonal COVID-19 vaccine, a third COVID-19 vaccine, and the next seasonal influenza vaccine prior to the interventions. This finding supports our floor effect suspicions and is in line with the previous studies, suggesting that vaccine hesitancy interventions might backfire when presented to highly vaccine hesitant people [27,28].”.

Comment 14: Flu - very helpful to see the flu data paralelled COVID data in some waysbut differed in other. This may well reflect different perceived context and hence differing importance between flua nd COVID vaccines, diff past experiences etcetc ie th emyriad other factors know impacts vaccien decsion making . This strengthen the paper

Comment 15: Discussion 527-530. Most studies of single interventions not change vaccine acceptance much - so not a big surprise here . Do wish had had 4th arm of stats + anecdote vs control

Response: Please see our response to Reviewer #2, Comment 12 above.

Comment 16: 576 -579 - yes context and expereince do matter ....

Comment 17: 591-595- as I could not read the anecdotes - don't know but likley correct

Response: Please see our response to Reviewer 2#, Comment 4.

Comment 18: 607 -608 might want to be more nuanced than this - stats or anecdotes indodulally did not appear to be effective in pop where not very postive torwards these vaccines.

Response: As the Reviewer suggested, we have modified the Conclusions section to highlight the fact that we did not test how these interventions had worked combined and in other samples, lines 806–817: “In the present study, the statistical and anecdotal vaccine hesitancy interventions turned out to be ineffective on their own at decreasing COVID-19 and influenza vaccine hesitancy, even when people received interventions in their preferred format. This finding suggests that tailoring vaccine hesitancy interventions according to format preference—as measured in the present study— might not increase the efficacy of the interventions, particularly in individuals who are highly hesitant. In fact, people who preferred anecdotes over statistics tended to react negatively toward both interventions, but even more so toward the anecdotal intervention. Taken together, this study highlights a need for caution when using anecdotal testimonies to persuade people who hold anti-vaccination attitudes and suggests that purely statistical vaccine hesitancy interventions might have a small edge over purely anecdotal interventions, due to the statistical interventions being slightly less likely to backfire.”.

Comment 19: 612-615- not sure can go that far- as never tested the two together - that has to eb one of the conlusions that that nees to be done

Response: To specify our final conclusion, we added that the statistical intervention might be preferable if statistical and anecdotal interventions are presented separately. Lines 813–817: “Taken together, this study highlights a need for caution when using anecdotal testimonies to persuade people who hold anti-vaccination attitudes and suggests that purely statistical vaccine hesitancy interventions might have a small edge over purely anecdotal interventions, due to the statistical interventions being slightly less likely to backfire.”.

Reviewer #3

Review for PONE-D-22-18819

Comment 1: This paper has two aims. First it develops a scale that aims to measure self-reported preferences for statistical or anecdotal information amongst vaccine hesitant individuals. Second it tests the interaction between format preferences (statistical or anecdotal information) and format type (statistical or anecdotal) on vaccine attitudes and intentions amongst vaccine deniers. The authors predicted that congruent information preference and intervention format (prefer statistical--informed statistical vs. preferred anecdotal--informed anecdotal) would boost vaccine intentions amongst vaccine hesitant participants. They found no support for their predictions. A non-predicted finding was that participants with preferences for anecdotes were more frustrated by the interventions regardless of format type (statistical, anecdotal) and found the interventions less helpful and relevant. Furthermore, in contrast to the predictions, the authors found that statistical information increased vaccine intentions of participants with higher preferences for anecdotal information. The topic of the paper is timely.

Theoretical background

It is unclear what the theoretical support for the predictions is. Why did the authors theoretically predict that interventions will be more effective in promoting vaccine intentions when the intervention format was congruent with the participants' format preference? And specifically, why did the authors predict this for vaccine denier? The paper would benefit from a clearer theoretical account. Without this it seems exploratory which also would be fine but then this needs to be clearly stated.

Response: We now describe the theories behind information formats and message tailoring in more detail in the Introduction section, lines 74–79: “Whereas statistical information is theorized to be persuasive because people perceive it as credible, verifiable, and generalizable [16], the persuasiveness of narratives has been attributed to their ability to reduce psychological reactance and counterarguing and to enable engagement in a storyline and identification with characters [17], while simultaneously being more easily processed than other types of formats [18].”, and lines 108–112: “Designing tailored interventions is believed to increase recipients’ attention to the message and by extension to make the message easier to remember. Tailored interventions are also expected to more often be perceived as relevant and to elicit more thorough elaboration of tailored messages compared to non-tailored ones [24].”. Moreover, we have also added that these studies are exploratory, and that we draw our hypotheses from previous research focusing on information format and message tailoring separately, lines 125–129: “Despite the fact that this study was exploratory in nature, we hypothesized based on previous information format and message tailoring research that the interventions would be more efficient in reducing vaccine hesitancy when the intervention format was congruent with the individuals’ format preference than when it was incongruent.”.

Comment 2: Methods

The preregistration OSF link presented in the paper did not work (I only found a OSF link for the data). For future submissions, I recommend that the authors are careful about including a link in their method section that works. It is crucial in judging the work. As a reviewer I need to understand whether the predictions made in the paper were a-priori or post-hoc, what inclusion and exclusion criteria were preregistered, whether the analyses were planned and what variables were used. I was unable to judge this as the link did not work.

Response: We wish to apologize to the Reviewer for failing to open the preregistration before the review process. The preregistration document is now unlocked and available for the public, and it can be found behind the same link that was provided in the previous version of the manuscript (osf.io/jqspe).

Comment 3: [Page 8, line 162 to 165] The power simulation presented in the paper seems undetailed. It would be helpful for the reader to include more details about what estimates were used and where they were based on.

Response: We have added a second supplemental materials document containing the entire power simulation process, including the R script used for the analysis. See the S1 File. Moreover, we briefly specify the process in the main text. Method section, line 218–224: “We manipulated the number of observations to find a large enough sample size for which a multiple regression model would reliably detect significant interaction terms between format preference and intervention groups. We found that a sample size of 600 participants per experiment would yield a statistical power of 0.87 for weak associations (three-point change on a scale from 0–100) between format preference and the efficacy of statistical and anecdotal interventions (see S1 File for an overview of the power simulation).”. We used centered means (mean = 0) for ease of simulation, and the standard deviations were rough estimates of what we expected the data to look like. Now, after having inspected the real observed standard deviations, we deleted the segment in the Discussion section, lines 660–664: “Important to note is that the statistical power of the experiments was sufficient to detect even small intervention effects and therefore the risk of Type II errors is low. Furthermore, the changes in vaccine attitudes and vaccination intentions were on average very small (close to zero on a 0–100 scale).”, as we found the differences between our estimates and the real observed values to be relatively big (estimated SD = 10, observed SD ≈ 20–30 ).

Comment 4: [Page 8, line 165 to 167] The exclusions of 843 participants is a very large part of the sample (43.4%). It would be important to understand whether this was preregistered and anticipated. But because the preregistration link did not work I am unable to judge this.

Response: Yes, this is stated in the preregistration.

Comment 5: Materials

It is great that the authors included examples of the materials that were used, but they are not readable in the current format. It would be important to include them in a higher quality so people can read and judge them. This is especially important given the null effect of the present study, which is informative for people who might want to test a similar intervention.

Response: We apologize for not presenting readable examples of the materials from the start. Fig 1 has been updated to an English translation, high-quality image of the COVID-19 intervention materials, and high-quality pictures and English translations of all intervention materials have now been added to the S2 File.

Comment 6: Discussion

The authors might wish to be more careful about their conclusions. They could for example be more careful in their wording and include the following suggestion or something along those lines (see **WORDING BELOW**):

[Page 30, line 526 to 528] “The present experiment consistently suggests that it is unlikely that tailoring vaccine hesitancy interventions according to format preferences **as measured in the present study with the present sample**,would enhance the efficacy of the interventions.

Response: As the Reviewer suggested, we have adjusted our conclusions to be more nuanced and specific. First, we removed a concluding statement from the beginning of the Discussion (lines 662–664), so that our conclusions are presented after discussing the results in light of previous research. We also made modification to the Conclusions section to highlight the limiting factors the Reviewer proposed, lines 806–817: “In the present study, the statistical and anecdotal vaccine hesitancy interventions turned out to be ineffective on their own at decreasing COVID-19 and influenza vaccine hesitancy, even when people received interventions in their preferred format. This finding suggests that tailoring vaccine hesitancy interventions according to format preference—as measured in the present study—might not increase the efficacy of the interventions, particularly in individuals who are highly hesitant. In fact, people who preferred anecdotes over statistics tended to react negatively toward both interventions, but even more so toward the anecdotal intervention. Taken together, this study highlights a need for caution when using anecdotal testimonies to persuade people who hold anti-vaccination attitudes and suggests that purely statistical vaccine hesitancy interventions might have a small edge over purely anecdotal interventions, due to the statistical interventions being slightly less likely to backfire.”.

Comment 7: I am wondering whether the intervention would be effective for uncertain participants. This group might benefit from tailored interventions. Is there a theoretical account that would suggest this?

Response: We too found this very interesting, which is why we conducted the post hoc analysis in which we separately examined the efficacy of the interventions on participants who were highly vaccine hesitant and on those who were instead uncertain (lines: 679–688). Also, we added further discussion about this topic to the Discussion section, lines 676–679: “On the other hand, a recent randomized controlled trial, investigating the effectiveness of different pro-vaccine messages on COVID-19 vaccine hesitancy, suggests the opposite, as the messages only reduced hesitancy in participants who were highly hesitant [41].”.

Comment 8: Best of luck with this research!

Academic Editor

Comment 1: When submitting your revision, we need you to address these additional requirements.

Response: The manuscript has been updated to match PLOS ONE’s style requirements.

Comment 2: 2. Please provide additional details regarding participant consent. In the ethics statement in the Methods and online submission information, please ensure that you have specified what type you obtained (for instance, written or verbal, and if verbal, how it was documented and witnessed). If your study included minors, state whether you obtained consent from parents or guardians. If the need for consent was waived by the ethics committee, please include this information.

Response: As requested, we have provided additional information about participant consent to the Ethics statement, lines 188–195: “Ethical permission for all studies was given by the Research Ethics Committee in Psychology and Logopedics of Åbo Akademi University. Before participating in the experiments, all participants were presented with information about the study and about the management of the data. Participants were then asked to indicate that they were at least 18 years old and that they had received enough information about the study to be able to give their informed consent to participate. All participants gave their written informed consent by ticking a box. Participants were also informed that participation was voluntary and that withdrawal from the study was possible at all times.”

Comment 3: 3. Thank you for stating the following financial disclosure:

“This project has received funding from the European Union’s Horizon 2020 research and innovation program under grant agreement No 964728 (JITSUVAX). KOM was funded by the Faculty of Social Sciences at the University of Turku (www.utu.fi/en/university/faculty-of-social-sciences). JKK and KOM received funding from the Strategic Research Council’s LITERACY program (Academy of Finland grant number: 335233). AS was funded by the Academy of Finland (grant number: 316004; www.aka.fi/en/). SL also acknowledges funding from the Humboldt Foundation Germany through a research award.

NO”

Response: As requested, the statement “The funders had no role in study design, data collection and analysis, decision to publish, or preparation of the manuscript.” is present in the cover letter.

Comment 4: 4. Please upload a new copy of Figure 1 as the detail is not clear. Please follow the link for more information: https://blogs.plos.org/plos/2019/06/looking-good-tips-for-creating-your-plos-figures-graphics/" https://blogs.plos.org/plos/2019/06/looking-good-tips-for-creating-your-plos-figures-graphics/

Response: Fig 1 has been updated to an English translation, high-quality image of the COVID-19 intervention materials, in line with PLOS ONE’s file and style requirements.

---

## [Decision Letter · Decision Letter 1]

1 Mar 2023

Tailoring interventions to suit self-reported format preference does not decrease vaccine hesitancy

PONE-D-22-18819R1

Dear Dr. Mäki,

We’re pleased to inform you that your manuscript has been judged scientifically suitable for publication and will be formally accepted for publication once it meets all outstanding technical requirements.

Kind regards,

Wojciech Trzebinski, Ph.D.

Academic Editor

PLOS ONE

Additional Editor Comments (optional):

Reviewers' comments:

Reviewer's Responses to Questions

**Comments to the Author**

1. If the authors have adequately addressed your comments raised in a previous round of review and you feel that this manuscript is now acceptable for publication, you may indicate that here to bypass the “Comments to the Author” section, enter your conflict of interest statement in the “Confidential to Editor” section, and submit your "Accept" recommendation.

Reviewer #2: All comments have been addressed

Reviewer #3: All comments have been addressed

2. Is the manuscript technically sound, and do the data support the conclusions?

Reviewer #2: Yes

Reviewer #3: Yes

3. Has the statistical analysis been performed appropriately and rigorously? 

Reviewer #2: Yes

Reviewer #3: Yes

4. Have the authors made all data underlying the findings in their manuscript fully available?

Reviewer #2: Yes

Reviewer #3: Yes

5. Is the manuscript presented in an intelligible fashion and written in standard English?

Reviewer #2: Yes

Reviewer #3: Yes

6. Review Comments to the Author

Reviewer #2: The study was never simple and hence the report was sometimes convoluted and hard to grasp. The changes in response to the reviewers' comments have made the paper much more accessible and the nuances more easily grasped. The report will add to the literature and further illustrates why simple solutions and easy leaps form previous data do not always pan out. The question challenged in this study needed an answer and sadly the negative finding may not be one many expected. Do hope the impact on actual uptake will be vetted at some point - not just intent

Reviewer #3: The authors addressed all my comments in a thorough manner. I have no further comments. Thank you very much.

7. PLOS authors have the option to publish the peer review history of their article (what does this mean?). If published, this will include your full peer review and any attached files.

Reviewer #2: No

Reviewer #3: No

---

## [Editor Report · Acceptance letter]

13 Mar 2023

PONE-D-22-18819R1 

Tailoring interventions to suit self-reported format preference does not decrease vaccine hesitancy 

Dear Dr. Mäki:

I'm pleased to inform you that your manuscript has been deemed suitable for publication in PLOS ONE. Congratulations! Your manuscript is now with our production department. 

Kind regards, 

on behalf of

Dr. Wojciech Trzebinski 

Academic Editor

PLOS ONE